# Language Model as Visual Explainer

**Xingyi Yang    Xinchao Wang**[*]
National University of Singapore
xyang@u.nus.edu, xinchao@nus.edu.sg

## Abstract

In this paper, we present Language Model as Visual Explainer (LVX), a systematic approach for interpreting the internal workings of vision models using a tree-structured linguistic explanation, without the need for model training. Central to our strategy is the collaboration between vision models and LLM to craft explanations. On one hand, the LLM is harnessed to delineate hierarchical visual attributes, while concurrently, a text-to-image API retrieves images that are most aligned with these textual concepts. By mapping the collected texts and images to the vision model's embedding space, we construct a hierarchy-structured visual embedding tree. This tree is dynamically pruned and grown by querying the LLM using language templates, tailoring the explanation to the model. Such a scheme allows us to seamlessly incorporate new attributes while eliminating undesired concepts based on the model's representations. When applied to testing samples, our method provides human-understandable explanations in the form of attribute-laden trees. Beyond explanation, we retrained the vision model by calibrating it on the generated concept hierarchy, allowing the model to incorporate the refined knowledge of visual attributes. To access the effectiveness of our approach, we introduce new benchmarks and conduct rigorous evaluations, demonstrating its plausibility, faithfulness, and stability.

## 1 Introduction

Unlocking the secrets of deep neural networks is akin to navigating through an intricate, ever-shifting maze, as the intricate decision flow within the networks is, in many cases, extremely difficult for humans to fully interpret. In this quest, extracting clear, understandable explanations from these perplexing mazes has become an imperative task.

While efforts have been made to explain computer vision models, these approaches often fall short of providing direct and human-understandable explanations. Standard techniques, such as attribution methods [46, 55, 90, 68, 1, 62, 65, 64], mechanical interpretability [22] and prototype analysis [10, 49], only highlight certain pixels or features that are deemed important by the model. As such, these methods often require the involvement of experts to verify or interpret the outputs for non-technical users. Natural language explanations [27, 7, 41, 35], on the other hand, present an attractive alternative, since the produced texts are better aligned with human understanding. Nevertheless, these approaches typically rely on labor-intensive and biased manual annotation of textual rationales for model training.

In this study, we attempt to explain AI decision in a human-understandable manner, for example, tree-structured language. We call this task *visual explanatory tree parsing*. To implement this, we present a systematic approach, Language Model as Visual Explainer (LVX), for interpreting vision models structured natural language, without model training.

---

[*]Corresponding author

38th Conference on Neural Information Processing Systems (NeurIPS 2024).

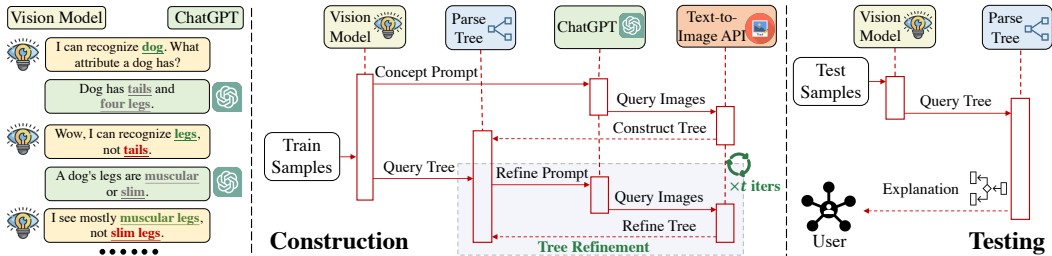

Figure 1: **General workflow of** LVX. **(Left)** A toy example that LLM interacts with vision model to examine its capability. **(Mid)** It combines vision, language, and visual-language APIs to create a parse tree for each visual model. **(Right)** In testing, embeddings navigate this tree, and the traversed path provides a personalized explanation for the model's prediction.

A key challenge is that vision models, trained solely on pixel data, inherently lack comprehension of textual concepts within an image. For example, if a model labels an image as a "*dog*", it is unclear whether it truly recognizes the features like the *wet nose* or *floppy ear*, or if it is merely making ungrounded guesses. To address this challenge, we link the vision model with a powerful external knowledge provider, to establish connections between textual attributes and image patterns. Specifically, we leverage large language models (LLM) such as ChatGPT and GPT4 as our knowledge providers, combining them with the visual recognition system. Figure 1 (Left) describes a toy case, where the LLM is interacts with the vision model to explore its capability boundaries. By doing so, we gain insights into the what visual attributes can be recognized by the model.

The pipeline of our approach is illustrated in Figure 1, which comprises two main stages, the *construction phase* and the *test phase*.

In the *construction phase*, our goal is to create an attribute tree for each category, partitioning the feature space of a visual model via LLM-defined hierarchy. We begin by extracting commonsense knowledge about each category and its visual attributes from LLMs using in-context prompting [45]. This information is naturally organized as a tree for better organization and clarity. Utilizing a text-to-image API, we gather corresponding images for each tree node. These images are subsequently inputted into the vision model to extract prototype embeddings, which are then mapped to the tree.

Once created, the tree is dynamically adjusted, based on the properties of the training set. Specifically, each embedding of the training sample is extracted by the vision model.Such embedding then navigates the parse tree based on their proximity to prototype embeddings. Infrequently visited nodes, representing attributes less recognizable, are pruned. Conversely, nodes often visited by the model indicate successful concept recognition. Those nodes are growned, as the LLM introduces more detailed concepts. Consequently, LVX yields human-understandable attribute trees that mirror the model's understanding of each concept.

In the *test phase*, we input a test sample into the model to extract its feature. The feature is then routed in the parse tree, by finding its nearest neighbors. The root-to-leaf path serves as a sample-specific rationale for the model, offering an explanation of how the model arrived at its decision.

To assess our method, we compiled new annotations and developed novel metrics. Subsequently, we test LVX on these self-collected real-world datasets to access its effectiveness.

Beyond interpretation, our study proposes to calibrate the vision model by utilizing the generated explanation results. The utilization of tree-structured explanations plays a key role in enhancing the model's performance, thereby facilitating more reliable and informed decision-making processes. Experimental results confirm the effectiveness of our method over existing interpretability techniques, highlighting its potential for advancing explainable AI.

To summarize, our main contributions are:

- The paper introduces a novel task, *visual explanatory tree parsing*, that interprets vision models using tree-structured language explanations.

- We introduce the Language Model as Visual Explainer (LVX) to carry out this task, without model training. The proposed LVX is the first dedicated approach to leverage LLM to explain the visual recognition system.

- Our study proposes utilizing the generated explanations to calibrate the vision model, leading to enhanced performance and improved reliability for decision-making.

- We introduce new benchmarks and metrics for a concise evaluation of the LVX method. These tools assess its plausibility, faithfulness, and stability in real-world datasets.

## 2 Problem Definition

We first define our specialized task, called **visual explanatory tree parsing**, which seeks to unravel the decision-making process of a vision model through a tree.

Let us consider the trained vision model $f$, defined as a function $f \colon \mathcal{X} \to \mathcal{Y}$, where $\mathcal{X}$ represents the input image space and $\mathcal{Y}$ denotes the output label space. In this study, our focus lies on the classification task, where $f = g \circ h$ is decomposed into a feature extractor $g$ and a linear classification head $h$. The output space is $\mathcal{Y} \in \mathbb{R}^n$, where $n$ signifies the number of classes. The model is trained on a labeled training set $D_{tr} = \{\mathbf{x}_j, y_j\}_{j=1}^{M}$, and would be evaluated a test set $D_{ts} = \{\mathbf{x}_j\}_{j=1}^{L}$.

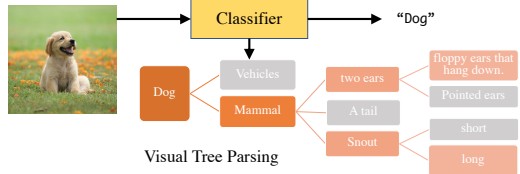

wmm

Figure 2: The illustration of visual explanatory tree parsing. Each input sample is interpreted as a parse tree to represent the model's logical process.

The ultimate objective of our problem is to generate an explanation $T$ for each model-input pair $(f, \mathbf{x})$ on the test set, illuminating the reasoning behind the model's prediction $\hat{y} = f(\mathbf{x})$. This unique explanation manifests as a tree of attributes, denoted as $T = (V, E)$, comprising a set of $N_v$ nodes $V = \{v_i\}_{i=1}^{N_v}$ and $N_e$ edges $E = \{e_i\}_{i=1}^{N_e}$. The root of the tree is the predicted category, $\hat{y}$, while each node $v_i$ encapsulates a specific attribute description of the object. These attributes are meticulously organized, progressing from the holistic to the granular, and from the general to the specific. Figure 2 provides an example of the parse tree.

Unlike existing approaches [53, 3] that explaining visual-language models [48, 47, 51, 86, 93], we address the more challenging scenario, on explaining vision models trained solely on pixel data. While some models can dissect and explain hierarchical clustering of feature embeddings [67, 77], they lack the ability to associate each node with a textual attribute. It is important to note that our explanations primarily focus on examining the properties of the established network, going beyond training vision model or visual-language model [2, 44] for reasoning hierarchy [20] and attributes [31] from the image. In other words, visual-language model, that tells the content in the image, can not explain the inner working inside another model. Notably, our approach achieves this objective *without supervision* and in *open-vocabulary* manner, without predefined explanations for model training.

## 3 Language Model as Visual Explainer

This section dives deeper into the details of LVX. At the heart of our approach is the interaction between the LLM and the vision model to construct the parsing tree. Subsequently, we establish a rule to route through these trees, enabling the creation of coherent text explanations.

### 3.1 Tree Construction via LLM

Before constructing our trees, let's take a moment to reflect how humans do this task. Typically, we already hold a hierarchy of concepts in our minds. When presented with visual stimuli, we instinctively compare the data to our existing knowledge tree, confirming the presence of distinct traits. We recognize familiar traits and, for unfamiliar ones, we expand our knowledge base. For example, when we think of a dog, we typically know that it has a *furry tail*. Upon observing a dog, we naturally check for the visibility of its tail. If we encounter a *hairless tail*, previously unknown

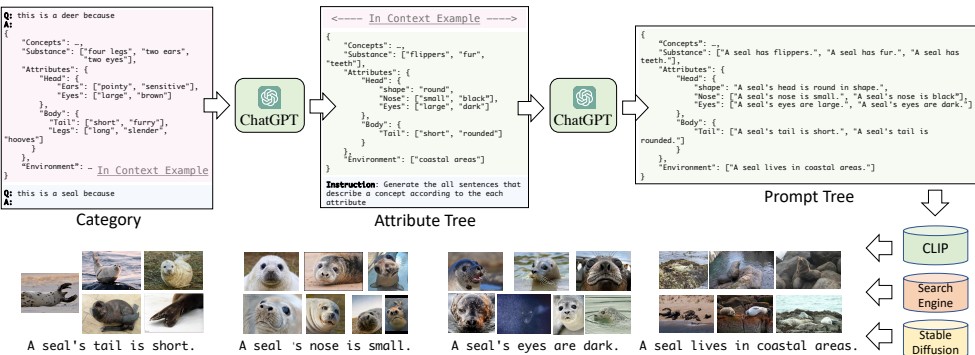

Figure 3: Crafting text-image pairs for visual concepts. Through in-context prompting, we extract knowledge from the LLM, yielding visual attributes for each category. These attributes guide the collection of text-image pairs that encapsulate the essence of each visual concept.

to us, we incorporate it into our knowledge base, ready to apply it to other dogs. This process is typically termed Predictive Coding Theory [15] in cognitive science.

Our LVX mirrors this methodology. We employ LLM as a "knowledge provider" to construct the initial conceptual tree. Subsequently, we navigate through the visual model's feature space to assess the prevalence of each node. If a specific attribute is rarely observed, we remove the corresponding nodes from the tree. Conversely, if the model consistently recognizes an attribute, we enrich the tree by integrating more nuanced, next-level concepts. This iterative process ensures the refinement and adaptation of the conceptual tree within our pipeline, which gives rise to our LVX.

**Generating Textual Descriptions for Visual Concepts.** We leverage a large language model (LLM) as our "commonsense knowledge provider" [42, 95] to generate textual descriptions of visual attributes corresponding to each category. The LLM acts as an external database, providing a rich source of diverse visual concept descriptions. The process is illustrated in Figure 3.

Formally, assume we have a set of category names, denoted as $C = \{c_i\}_{i=1}^n$, where $i$ represents the class index. For each of these classes, we prompt an LLM $L$ to produce visual attribute tree. We represent these attributes as $d_i = L(c_i, \mathcal{P})$, where $d_i$ is a nested JSON text containing textual descriptions associated with class $c_i$. To help generate $d_i$, we use example input-output pairs, $\mathcal{P}$, as in-context prompts. The process unfolds in two stages:

- **Initial Attribute Generation**: We initially generate keywords that embody the attributes of each class. This prompt follows a predefined template that instructs the LLM to elaborate on the attributes of a visual object. The template is phrased as ("This is a <CLSNAME> because"). The output JSON contains four primary nodes: Concepts, Substances, Attributes, and Environments. As such, the LLM is prompted to return the attributes of that concept. Note that the initial attributes tree may not accurately represent the model; refinements will be made in the refinement stage.

- **Description Composition**: Next, we guide the LLM to create descriptions based on these attributes. Again we showcase an in-context example and instruct the model to output ("Generate sentences that describe a concept according to each attribute.").

Once the LLM generates the structured attributes $d_i$, we parse them into an initial tree, represented as $T_i^{(0)} = (V_i^{(0)}, E_i^{(0)})$, using the key-value pairs of the JSON text. Those generated JSON tree is then utilized to query images corresponding to each factor.

**Visual Embeddings Tree from Retrieved Images.** In order to enable the vision model to understand attributes generated by the LLM, we employ a two-step approach. The primary step involves the conversion of textual descriptions, outputted by the LLM, into images. Then, these images are deployed to investigate the feature region that symbolizes specific attributes within the model.

The transition from linguistic elements to images is facilitated by the use of arbitrary text-to-image API. This instrumental API enables the generation of novel images or retrieval of existing images that bear strong relevance to the corresponding textual descriptions. An initial parse tree node, denoted by

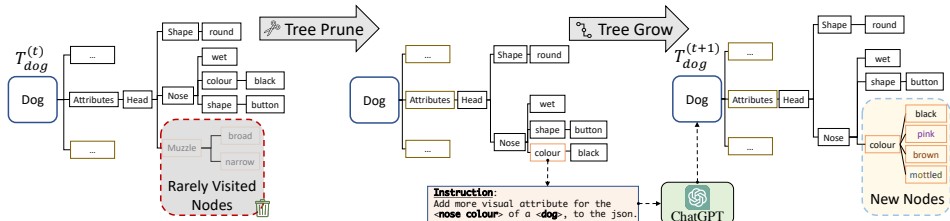

Figure 4: Tree refinement by traversing the embedding tree and querying the LLM model.

$v$, containing a textual attribute, is inputted into the API to yield a corresponding set of $K$ support images, represented as $\{\widetilde{\mathbf{x}}_i\}_{i=1}^K = \texttt{T2I}(v)$. The value of $K$ is confined to a moderately small range, typically between 5 to 30. The full information of the collected dataset will be introduced in Section 4.

Our research incorporates the use of search engines such as Bing, or text-to-image diffusion models like Stable-Diffusion [56], to derive images that correspond accurately to the provided attributes.

Following this, the images are presented to the visual model to extract their respective embeddings, represented as $\mathbf{p}_i = g(\widetilde{\mathbf{x}}_i)$. As such, each tree node contains a set of support visual features $P = \{\mathbf{p}_k\}_{k=1}^K$. This procedure allows for the construction of an embedding tree, consisting of paired text and visual features. These pairs are arranged in a tree structure prescribed by the LLM. It is important to note that the collected images are not employed in training the model. Instead, they serve as a support set to assist the model in understanding and representing the disentangled attributes effectively. As such, the visual model uses these embeddings as a map to navigate through the vast feature space, carving out territories of attributes, and laying down the groundwork for further exploration and explanation of a particular input.

**Tree Refinement Via Refine Prompt.** Upon construction, the parse tree structure is refined to better align with the model's feature spaces. This stage, termed *Tree Refinement*, is achieved through passing training data as a query to traverse the tree. Nodes that are seldom visited indicate that the model infrequently recognizes their associated attributes. Therefore, we propose a pruning mechanism that selectively eliminates these attributes, streamlining the tree structure. For nodes that frequently appear during the traversal, we further grow the tree by introducing additional or more detailed attributes, enriching the overall context and depth of the tree. The procedure is demonstrated in Figure 4.

Initially, we treat the original training samples, denoted as $(\mathbf{x}_j, y_j) \in D_{tr}$, as our query set. Each sample is passed to the visual model to extract a feature, represented as $\mathbf{q}_j = g(\mathbf{x}_j)$.

Next, the extracted feature traverses the $y_j$-corresponding tree. Its aim is to locate the closest semantic neighbors among the tree nodes. We define a distance metric between $\mathbf{q}_j$ to support set $P$ as the point-to-set distance $D(\mathbf{q}_j, P)$. This metric represents the greatest lower bound of the set of distances from $\mathbf{q}_j$ to prototypes in $P$. It is resilient to outliers and effectively suppresses non-maximum nodes.

$$D(\mathbf{q}_j, P) = \inf\{d(\mathbf{q}_j, \mathbf{p}) | \mathbf{p} \in P\} \tag{1}$$

In our paper, similar to [10, 58], we set $d(\mathbf{q}, \mathbf{p}) = -\log(1 + \frac{1}{||\mathbf{q}-\mathbf{p}||^2})^2$. It emphasizes close points while moderating the impact of larger distances. Following this, we employ a Depth-First Search (DFS) algorithm to locate the tree node closest to the query point $\mathbf{q}_j$. After finding this node, each training point $(\mathbf{x}_j, y_j)$ is assigned to a specific node of the tree. Subsequently, we count the number of samples assigned to a particular node $v^*$, using the following formula:

$$C_{v^*} = \sum_{j=1}^M \mathbb{1}\{v^* = \operatorname*{argmin}_{v \in V_{y_j}^{(0)}} D(\mathbf{q}_j, P_v)\} \tag{2}$$

In this formula, $\mathbb{1}$ is the indicator function and $P_v$ denotes the support feature for node $v$. Following this, we rank each node based on the sample counter, which results in two operations to update the tree architecture $T_i^{(t+1)} = \texttt{Grow}(\texttt{Prune}(T_i^{(t)}))$, where $t$ stands as the iteration number

• **Tree Pruning**. Nodes with the least visits are pruned from the tree, along with their child nodes.

---

[2]In practice, we implement $d(\mathbf{q}, \mathbf{p}) = -\log(\frac{||\mathbf{q}-\mathbf{p}||^2+1}{||\mathbf{q}-\mathbf{p}||^2+\epsilon})$, incorporating $\epsilon > 0$ to ensure numerical stability.

- **Tree Growing**. For the top-ranked node, we construct a new inquiry to prompt the LLM to generate attributes with finer granularity. The inquiry is constructed with an instruction template `"Add visual attributes for the <NODENAME> of a <CLASSNAME>, to the json"`.

- **Common Node Discrimination**. In cases where different categories share common nodes (e.g. "human" and "dog" both have "ear"), we execute a targeted growth step aimed at distinguishing between these shared elements. To achieve this differentiation, we utilize a contrasting question posed to the LLM `"The <NODENAME> of <CLASSNAME1> is different from <CLASSNAME2> because"`.

The revised concept tree generated by the LLM provides a comprehensive and detailed representation of the visual attribute. To refine the attribute further, we employ an iterative procedure that involves image retrieval and the extraction of visual embeddings, as illustrated in Figure 1. This iterative process enhances the parse tree by incorporating new elements. As each new element is introduced, the attribute areas within the feature space become increasingly refined, leading to improved interpretability. In our experiment, we performed five rounds of tree refinement.

## 3.2   Routing in the Tree

Once the tree is established, the model predicts the class of a new test sample $\mathbf{x}'$ and provides an explanation for this decision by finding the top-k nearest neighbor nodes.

Specifically, the model predicts the category $\hat{y}$ for the test instance $\mathbf{x}'$ as $\hat{y} = f(\mathbf{x}')$. The extracted image feature $\mathbf{q}'$ corresponding to $\mathbf{x}'$ is routed through the tree. Starting from the root, the tree is traversed to select the top-k nearest neighbor nodes $\{v_i\}_{i=1}^k$ based on the smallest $D(\mathbf{q}', P_{v_i})$ values, representing the highest semantic similarity between $\mathbf{q}'$ and the visual features in the tree's nodes. The paths from the root to the selected nodes are merged to construct the explanatory tree $T$ for the model's prediction.

This parse tree structure reveals the sequence of visual attributes that influenced the model's classification of $\mathbf{x}'$ as $\hat{y}$. It facilitates the creation of precise, tree-structured justifications for these predictions. Importantly, the routing process involves only a few feature similarity computations per node and does not require queries to the large language model, resulting in exceptionally fast computation.

## 3.3   Calibrating through Explaining

The created parse tree offers a two-fold advantage. Not only does it illustrates the logic of a specific prediction, but it also serves as a by-product to refine the model's predictions by introducing hierarchical regularization for learned representation. Our goal is to use the parse tree as pseudo-labels, embedding this hierarchical knowledge into the model.

To operationalize this, we employ a hierarchical multi-label contrastive loss (HiMulCon) [92], denoted as $\mathcal{L}_{HMC}$, to fine-tune the pre-trained neural network. This approach enhances the model by infusing structured explanations into the learning process, thus enriching the representation.

Specifically, we apply the LVX on all training samples. The explanatory path $\hat{T}_j$ provides a hierarchical annotation for each training sample $\mathbf{x}_j$. The model is trained with both the cross-entropy loss $\mathcal{L}_{CE}$ and $\mathcal{L}_{HMC}$ as follows:

$$\min \sum_{j=1}^M \mathcal{L}_{CE}\Big(f(\mathbf{x}_j), y_j\Big) + \lambda \mathcal{L}_{HMC}\Big(g(\mathbf{x}_j), \hat{T}_j\Big) \tag{3}$$

Here, $\lambda$ is a weighting coefficient. The explanation $\hat{T}_j$ is updated every 10 training epochs to ensure its alignment with the network's evolving parameters and learning progress. Notably, the support set isn't used in model training, maintaining a fair comparison with the baselines.

# 4   Experiment

This section offers an in-depth exploration of our evaluation process for the proposed LVX framework and explains how it can be utilized to gain insights into the behavior of a trained visual recognition model, potentially leading to performance and transparency improvements.

Table 1: Data annotation statistics. The ∗ indicates the number of video frames. We compare the statistics of category, attributes, image and tree depth across different explanatory datasets. Our dataset stands out as the first hierarchical dataset, offering a wide range of attributes.

| Dataset Name | No. Class | No. Attr | No. Images | Avg. Tree Depth | Rationales | Hierarchy | Validation Only |
|---|---|---|---|---|---|---|---|
| AWA2 [81] | 50 | 85 | 37,322 | N/A | ✓ | ✗ | ✗ |
| CUB [76] | 200 | N/A | 11,788 | N/A | ✓ | ✗ | ✗ |
| BDD-X [35] | 906 | 1,668 | 26,000* | N/A | ✓ | ✗ | ✗ |
| VAW [52] | N/A | 650 | 72,274 | N/A | ✗ | ✗ | ✗ |
| COCO Attr [50] | 29 | 196 | 180,000 | N/A | ✗ | ✗ | ✗ |
| DR-CIFAR-10 [47] | 10 | 63 | 2,201 | N/A | ✓ | ✗ | ✗ |
| DR-CIFAR-100 [47] | 100 | 540 | 18,318 | N/A | ✓ | ✗ | ✗ |
| DR-ImageNet [47] | 1,000 | 5,810 | 271,016 | N/A | ✓ | ✗ | ✗ |
| H-CIFAR-10 | 10 | 289 | 10,000 | 4.3 | ✓ | ✓ | ✓ |
| H-CIFAR-100 | 100 | 2,359 | 10,000 | 4.5 | ✓ | ✓ | ✓ |
| H-ImageNet | 1,000 | 26,928 | 50,000 | 4.8 | ✓ | ✓ | ✓ |

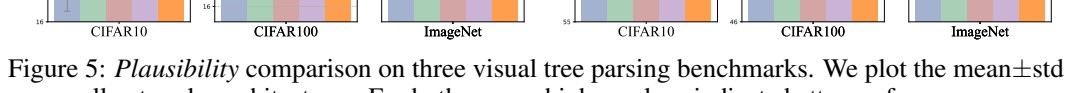

Figure 5: *Plausibility* comparison on three visual tree parsing benchmarks. We plot the mean±std across all networks architectures. For both scores, higher values indicate better performance.

## 4.1 Experimental Setup

**Data Annotation and Collection.** To assess explanation plausibility, data must include human annotations. Currently, no large-scale vision dataset with hierarchical annotations is available to facilitate reasoning for visual predictions. To address this, we developed annotations for three recognized benchmarks: CIFAR10, CIFAR100 [39], and ImageNet [57], termed as `H-CIFAR10`, `H-CIFAR100`, and `H-ImageNet`. These annotations, detailed in Table 8, serve as ground truth for model evaluation, highlighting our dataset's unique support for hierarchical attributes and diverse visual concepts. Note that, we evaluate on hierarchical datasets only, as our method is specifically designed for structured explanations.

As an additional outcome of our framework, we have gathered three support sets to facilitate model explanation. In these datasets, each attribute generated by the LLM corresponds to a collection of images that showcase the specified visual concepts. These images are either retrieved from Bing search engine [3] using attributes as queries or are generated using Stable-diffusion. We subsequently filter the mismatched pairs with the CLIP model, with the threshold of 0.5. Due to the page limit, extensive details on data collection, false positive removal, limitations, and additional evaluation of user study and on medical data, such as X-ray diagnoses, are available in the supplementary material.

**Evaluation Metrics.** In this paper, we evaluate the quality of our explanation from three perspectives: *Plausibility*, *Faithfulness* and *Stability*.

- **Plausibility** measures how reasonable the machine explanation is compared to the human explanation. We measure this by the graph distance between the predicted and ground-truth trees, using two metrics: Maximum Common Subgraph (MCS) [54, 33], and Tree Kernels (TK) [71]. We calculate their normalized scores respectively. Specifically, given a predicted tree $T_{pred}$ and the ground-truth $T_{gt}$, the MCS score is computed as $\frac{|MCS| \times 100}{\sqrt{|T_{pred}||T_{gt}|}}$, and the TK score is computed as $\frac{TK(T_{pred}, T_{gt}) \times 100}{\sqrt{TK(T_{pred}, T_{pred})TK(T_{gt}, T_{gt})}}$. Here, $|\cdot|$ represents the number of nodes in a tree, and $TK(\cdot, \cdot)$ denotes the unnormalized TK score. We report the average score across all validation samples.

- **Faithfulness** states that the explanations should reflect the inner working of the model. We introduce Model-induced Sample-Concept Distance (MSCD) to evaluate this, calculated as the

---
[3]https://www.bing.com/images/

Table 2: *Stability* comparison in CIFAR10 under input perturbations.

| Method | Network | Clean | | Gaussian $(\sigma = 0.05)$ | | Gaussian $(\sigma = 0.1)$ | | Cutout $(n_{\text{holes}} = 1)$ | |
|---|---|---|---|---|---|---|---|---|---|
| | | MCS | TK | MCS | TK | MCS | TK | MCS | TK |
| TrDec | RN-18 | 100 | 100 | 65.3 | 86.4 | 56.2 | 82.5 | 65.4 | 86.0 |
| LVX | RN-18 | 100 | 100 | **69.7** | **90.8** | **62.1** | **86.5** | **68.1** | **88.3** |
| TrDec | RN-50 | 100 | 100 | 68.3 | 88.5 | 59.3 | 84.2 | 66.2 | 86.9 |
| LVX | RN-50 | 100 | 100 | **71.9** | **92.1** | **65.6** | **88.3** | **69.3** | **90.1** |

Table 3: *Faithfulness* comparison by computing the MSCD score. Smaller the better.

| Network | CIFAR-10 | | | CIFAR-100 | | | ImageNet | | |
|---|---|---|---|---|---|---|---|---|---|
| | TrDec | SubTree | LVX | TrDec | SubTree | LVX | TrDec | SubTree | LVX |
| RN-18 | -0.224 | -0.393 | **-0.971** | -0.246 | -0.446 | **-0.574** | -0.298 | -0.548 | **-0.730** |
| RN-50 | -0.236 | -0.430 | **-1.329** | -0.256 | -0.500 | **-1.170** | -0.317 | -0.588 | **-1.186** |
| ViT-S 16 | -0.244 | -0.467 | **-1.677** | -0.266 | -0.527 | **-1.073** | -0.330 | -0.626 | **-1.792** |

average of point-to-set distances $\frac{1}{N_v} \sum_{v \in V} D(\mathbf{q}_j, P_v)$ between all test samples and tree nodes, reflecting the alignment between generated explanation and model's internal logic. The concept is simple: if the explanation tree aligns with the model's internal representation, the MSCD is minimized, indicating high faithfulness.

- **Stability** evaluates the resilience of the explanation graph to minor input variation, expecting minimal variations in explanations. The MCS/TK metrics are used to assess stability by comparing explanations derived from clean and slightly modified inputs. We include 3 perturbations, including Gaussian additive noise with $\sigma \in \{0.05, 0.1\}$ and Cutout [18] augmentation.

**Baselines.** We construct three baselines for comparisons: `Constant`, using the full category template tree; `Random`, which selects a subtree randomly from the template; and `Subtree`, choosing the most common subtree in the test set for explanations. Additionally, we consider `TrDec` Baseline [79], a strategy utilizing a tree-topology RNN decoder on top of image encoder. Given the absence of hierarchical annotations, the CLIP model verifies nodes in the template trees, serving as pseudo-labels for training. We only update the decoder parameters for interpretation purposes. These models provide a basic comparison for the performance of `LVX`. More details are in the appendix.

For classification performance, we compare `LVX`-calibrated model with neural-tree based solutions, including a Decision Tree (DT) trained on the neural network's final layer, DNDF [38], and NBDT [77].

**Models to be Explained.** Our experiments cover a wide range of neural networks, including various convolutional neural networks (CNN) and transformers. These models consist of VGG [66], ResNet [26], DenseNet [29], GoogLeNet [72], Inceptionv3 [73], MobileNet-v2 [59], and Vision Transformer (ViT) [19]. In total, we utilize 12 networks for CIFAR-10, 11 networks for CIFAR-100, and 8 networks for ImageNet. For each model, we perform the tree refinement for 5 iterations.

**Calibration Model Training.** As described in Section 3.3, we finetune the pre-trained neural networks with the hierarchical contrastive loss based on the explanatory results. The model is optimized with SGD for 50 epochs on the training sample, with an initial learning rate in $\{0.001, 0.01, 0.03\}$ and a momentum term of 0.9. The weighting factor is set to 0.1. We compare the calibrated models with the original ones in terms of accuracy and explanation faithfulness.

## 4.2 LLM helps Visual Interprebility

**Plausibility Results.** We evaluated `LVX` against human annotations across three datasets, using different architectures, and calculating MCS and TK scores. The results, shown in Figure 5, reveal `LVX` outperforms baselines, providing superior explanations. Notably, `TrDec`, even when trained on CLIP induced labels, fails to generate valid attributes in deeper tree layers—a prevalent issue in long sequence and structure generation tasks. Meanwhile, `SubTree` lacks adaptability in its explanations, leading to lower scores. More insights are mentioned in the appendix.

**Faithfulness Results.** We present the MSCD scores for ResNet-18(RN-18), ResNet-50(RN-50), and ViT-S, contrasting them with `SubTree` and `TrDec` in Table 3. Thanks to the incorporation of tree refinement that explicitly minimizes MSCD, our `LVX` method consistently surpasses benchmarks, demonstrating lowest MSCD values, indicating its enhanced alignment with model reasoning.

**Stability Results.** The stability of our model against minor input perturbations on the CIFAR-10 dataset is showcased in Table 2, where MCS/TK are computed. The "Clean" serves as the oracle baseline. Our method, demonstrating robustness to input variations, retains consistent explanation results (MCS>60, TK>85). In contrast, `TrDec`, dependent on an RNN-parameterized decoder, exhibits higher sensitivity to feature variations.

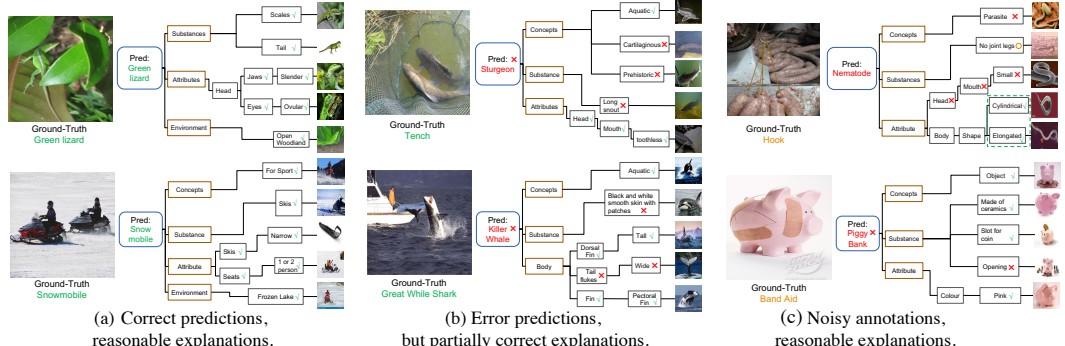

(a) Correct predictions,
reasonable explanations.

(b) Error predictions,
but partially correct explanations.

(c) Noisy annotations,
reasonable explanations.

Figure 6: Explanation visualization for ViT-B on ImageNet-1K. ✓ and × means that the node is aligned or misaligned with the image. Zoom in for better view.

| Method | Network | Expl. | CIFAR10 | CIFAR100 | ImageNet |
|---|---|---|---|---|---|
| NN | ResNet18 | × | 94.97% | 75.92% | 69.76% |
| DT | ResNet18 | ✓ | 93.97% | 64.45% | 63.45% |
| DNDF | ResNet18 | ✓ | 94.32% | 67.18% | N/A |
| NBDT | ResNet18 | ✓ | 94.82% | 77.09% | 65.27% |
| LVX (Ours) | ResNet18 | ✓ | **95.14%** | **77.33%** | **70.28%** |

Table 4: Performance comparison of neural decision tree-based methods. *Expl.* stands for whether the prediction is explainable.

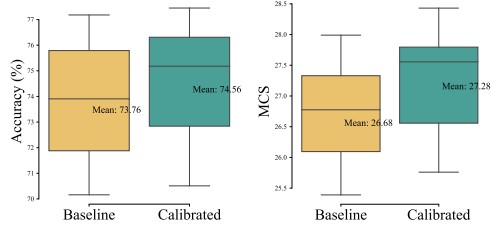

Table 5: Performance and interpretability comparison with/without model calibration on CIFAR-100. Higher MCS means better.

**Model and Data Diagnosis with Explanation.** We visualize the sample explanatory parse tree on ImageNet validation set induced by ViT-B in Figure 6. The explanations fall into three categories: (1) correct predictions with explanations, (2) incorrect predictions with explanations, and (3) noisy label predictions with explanations. We've also displayed the 5 nearest neighbor node for each case.

What's remarkable about LVX is that, even when the model's prediction is wrong, it can identify correct attributes. For instance, in a case where a "white shark" was misidentified as a "killer whale" (b-Row 2), LVX correctly identified "fins", a shared attribute of both species. Moreover, the misrecognition of the attribute "wide tail flukes" indicates a potential error in the model, that could be later addressed to enhance its performance.

Surprisingly, LVX is able to identify certain noisy labels in the data, as shown in c-Row 2. In such cases, even experienced human observers might struggle to decide whether a "pig bank with band" should be classified "piggy bank" or "band aid". It again underscores the superior capabilities of our LVX system in diagnosing the errors beyond model, but also within the data itself.

**Calibration Enhances Interpretability and Performance.** Our approach involves fine-tuning a pre-trained model with the loss function outlined in Section 3.3, using parsed explanatory trees to improve model performance. Table 4 compares the classification performance of our model with that of other neural tree methods. Our model clearly outperforms the rest.

Neural tree models often face challenges in balancing interpretability with performance. In contrast, LVX achieves strong performance without relying on a strict decision tree. Instead, decisions are handled by the neural network, with concepts guided by the LLM through Equation 3. This approach enhances the model's ability to disentangle visual concepts while preserving high performance.

In addition, we compared the quality of the generated parsed tree with or without calibration, in Figure 5. The calibration process not only improved model performance, but also led to more precise tree predictions, indicating enhanced interpretability. We also test the calibrated model on OOD evaluations in Appendix, where we observe notable improvements.

## 5 Ablation Study and Analysis

In this section, we present an ablation study on the refinement stage of LVX. We also apply the method to different neural networks to observe variations in model's behavior.

Table 6: Performance comparison on CIFAR-10 and CIFAR-100 with and without refinement. Higher MCS and lower MSCD indicate better performance.

| Network | Method | CIFAR-10 | | CIFAR-100 | |
|---|---|---|---|---|---|
| | | MCS | MSCD | MCS | MSCD |
| ResNet-18 | w/o Refine | 27.73 | -0.645 | 23.18 | -0.432 |
| | LVX | **30.24** | **-0.971** | **25.10** | **-0.574** |
| ResNet-50 | w/o Refine | 28.09 | -0.822 | 23.44 | -0.698 |
| | LVX | **31.09** | **-1.329** | **26.90** | **-1.170** |

**Ablation 1: No Refinement.** To study the impact of refinement stage, we present a baseline called *w/o Refine*. In this setup, the initial tree generated by LLMs is kept fixed. We evaluate the method using the MSCD for faithfulness and MCS for plausibility on the CIFAR-10 and CIFAR-100 datasets.

The results show in Table 6 that incorporating image model feedback indeed improves tree alignment with the classifier's internal representation, as reflected in higher MCS scores. The refined trees also better match human-annotations.

**Ablation 2: Refinement Criteria.** In our original method, tree refinement is based on feature similarity to the training set. To explore an alternative, we use *average activation magnitude* on generated data as the criterion for concept familiarity. Concepts with activation magnitudes $\leq \eta$ are pruned. This method, referred to as *ActMag*, is evaluated on CIFAR-10. We report the MCS, MSCD for performance, and average tree depth as an indicator of tree complexity.

Table 7 shows that feature similarity achieves better results than *ActMag*. Specifically, setting a threshold is challenging for *ActMag*, leading shallow trees ($\eta = 0.3$) or too deep ones ($\eta = 0.01$).

Table 7: Performance comparison on CIFAR-10 with different refinement criteria.

| Network | LVX | | | ActMag($\eta = 0.01$) | | | ActMag($\eta = 0.1$) | | | ActMag($\eta = 0.3$) | | |
|---|---|---|---|---|---|---|---|---|---|---|---|---|
| | MCS | MSCD | Depth | MCS | MSCD | Depth | MCS | MSCD | Depth | MCS | MSCD | Depth |
| ResNet-50 | **31.1** | **-1.3** | 4.2 | 23.4 | -0.3 | 6.0 | 26.9 | -0.8 | 3.7 | 25.3 | -0.5 | 1.4 |
| ViT-S | **31.9** | **-1.7** | 4.3 | 24.2 | -0.4 | 6.2 | 27.4 | -0.9 | 3.3 | 26.1 | -0.6 | 1.8 |

**Analysis: CNN vs. Transformer.** We use our LVX to compare CNN and Transformer models and identify which concepts they miss. We compared ConvNeXt-T (CNN) and DeiT-B (Transformer) on 26,928 concepts we collected on ImageNet, from sub-categories of *Concepts*, *Substances*, *Attributes*, and *Environments*. We measured accuracy across 4 sub-categories and tree depths.

Results show that ConvNeXt-T is better at local patterns (Attributes, Substances), while DeiT-B perform better on Environments which needs global semantics. Additionally, DeiT-B is more accurate at shallow depths, whereas ConvNeXt-T performs better at deeper levels. These findings aligns with earlier research showing that CNN are biased towards textures over shape [23, 84].

# 6 Conclusion

In this study, we introduced LVX, an approach for interpreting vision models using tree-structured language explanations without hierarchical annotations. LVX leverages large language models to connect visual attributes with image features, generating comprehensive explanations. We refined attribute parse trees based on the model's recognition capabilities, creating human-understandable descriptions. Test samples were routed through the parse tree to generate sample-specific rationales. LVX demonstrated effectiveness in interpreting vision models, offering potential for model calibration. Our contributions include proposing LVX as the first approach to leverage language models for explaining the visual recognition system. We hope this study potentially advances interpretable AI and deepens our understanding of neural networks.

# Acknowledgement

This project is supported by the Ministry of Education, Singapore, under its Academic Research Fund Tier 2 (Award Number: MOE-T2EP20122-0006), and the National Research Foundation, Singapore, under its AI Singapore Programme (AISG Award No: AISG2-RP-2021-023).

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

## A    Appendix / supplemental material

This document presents supplementary experiments and information regarding our proposed LVX framework. In Section C, we provide an overview of the algorithm pipeline for LVX. In Section D, we detail the process of data collection and its subsequent analysis. In additionally, Section E provides the user study results; Section F shows the calibrated training further improve the OOD performance. Section G showcases the additional experimental results obtained from a specialized application on X-Ray diagnosis. Furthermore, in Section H, we demonstrate the explanation results achieved using self-supervised models. We also provide the raw experimental values in Section I. Finally, we outline the experimental setup, metric definitions, and dataset collection protocols.

## B    Related Work

**Neural Tree.** Neural Trees (NTs) intend to harmonize the performance of Neural Networks (NNs) and interpretability of Decision Trees (DTs) [17, 21, 60, 96, 14] within a unified model. They evolved from mimicking NNs with DTs [17, 21, 60, 96, 14] to becoming inherently interpretable tree-structured networks, adapting their structure via gradient descent [70, 94, 32, 85, 75, 38]. Neural-Backed Decision Trees (NBDTs) [77] use a trained NN as a feature extractor, replacing its final layer with a decision tree. Our model builds on these advances to create a hierarchical tree from a pre-trained NN and provides *post-hoc* explanations without additional training, which increases interpretability and potentially enhances performance.

**Prototype-based Explainable Model.** Prototype models use representative training data points to symbolize classes or outcomes [16, 30, 37]. Revived in deep learning and few-shot learning [69, 82], they justify decisions by comparing new instances to key examples [10, 88, 40]. Recent work has developed this approach through hierarchical and local prototypes [49, 34, 74]. However, the prototypes serve as an indirect explanation for model's prediction, necessitating further human justification. Our LVX addresses this by assigning semantic roles to prototypes through LLM, turning them from simple similarity points to data points with clear definitions, thereby enhancing explainability.

**Composing Foundation Models.** Model composition involves merging machine learning models to address tasks, often using modular models for sub-tasks [5, 28, 4, 83], restructured by a symbolic executor [89]. Recently, Language Learning Models (LLMs) have been used as central controllers, guiding the logic of existing models [63, 25, 87, 43] or APIs [61], with language as a universal interface [91]. However, composing language model with non-language ones lacks a unified interface for bilateral communication. In this study, we propose the use of a text-to-image API as a medium to enable the language model to share its knowledge with the visual task. This allows the vision model to benefit from the linguistic context and knowledge hierarchy, thereby enhancing its transparency.

## C    Pseudo-code for LVX

In this section, we present the pseudocode for the LVX framework, encompassing both the *construction stage* and the *test stage*. The algorithmic pipelines are outlined in Algorithm 1 and Algorithm 2.

The category-level tree construction pipeline, as demonstrated in Algorithm 1, involves an iterative process that utilizes a large language model (LLM), like ChatGPT. This process allows us to construct an attribute tree for each category. It begins by generating prompts based on the category name and using them to gather attribute information. This forms the initial tree structure. Support images are collected using a text-to-image API, and their visual features are associated with the corresponding tree nodes. The process iterates until the maximum run, continuously refining the attribute tree for corresponding category.

During the test stage, as outlined in Algorithm 2, the test samples undergo a traversal process within the constructed category-level trees. The goal is to locate the subtree that best aligns with the correct

---

**Algorithm 1** Language Model as Visual Explainer (LVX)-Construction

---

**Input:** Vision model $f = \circ h$, a large language model $L$, a text-to-image API T2I, a training set $D_{tr} = \{\mathbf{x}_i, y_i\}_{i=1}^M$, class names $C = \{c_i\}_{i=1}^n$ and a concept prompt tree input-output example $\mathcal{P}$.
**Output:** An explanatory tree $T_i$ for each category $c_i$.
1: // Construct the initial Parse Tree
2: **for** $i = 1$ **to** $n$ **do**
3:     In-context Prompt LLM: $d_i = L(c_i, \mathcal{P})$.
4:     Parse $d_i$ into an initial tree $T_i^{(0)} = \{V_i^{(0)}, E_i^{(0)}\}$.
5:     Collect support images from text-to-image API: $\{\widetilde{\mathbf{x}}_i\}_{i=1}^K = \texttt{T2I}(v)$, where $v \in V_i^{(0)}$.
6:     Extract features in each tree node: $P_v = \{\mathbf{p}_i\}_{i=1}^K = \{g(\widetilde{\mathbf{x}}_i) | \widetilde{\mathbf{x}}_i \in \{\widetilde{\mathbf{x}}_i\}_{i=1}^K\}$.
7: **end for**
8: // Parse Tree Refinement
9: **for** $t = 0$ **to** $t_{\max}$ **do**
10:     **for** $j = 1$ **to** $M$ **do**
11:         Extract feature for training data: $\mathbf{q}_j = g(\widetilde{\mathbf{x}}_j)$.
12:         Assign training data to a tree node: $v^* = \operatorname{argmin}_{v \in V_{y_j}^{(t)}} D(\mathbf{q}_j, P_v)$.

13:     **end for**
14:     Count the number of samples for each node: $C_{v^*} = \sum_{j=1}^M \mathbb{1}\{v^* = \operatorname{argmin}_{v \in V_{y_j}^{(0)}} D(\mathbf{q}_j, P_v)\}$

15:     Prune the least visited node: $T_i^{(t)} = \texttt{Prune}(T_i^{(t)})$.
16:     Grow the most visited node: $T_i^{(t+1)} = \texttt{Grow}(T_i^{(t)})$.
17:     Collect support images from text-to-image API: $\{\widetilde{\mathbf{x}}_i\}_{i=1}^K = \texttt{T2I}(v)$, where $v \in V_i^{(t+1)}$.
18:     Extract features in new tree node: $P_v = \{\mathbf{p}_i\}_{i=1}^K = \{g(\widetilde{\mathbf{x}}_i) | \widetilde{\mathbf{x}}_i \in \{\widetilde{\mathbf{x}}_i\}_{i=1}^K\}$.
19: **end for**
20: **return** $T_i^{(t_{\max})}$ as $T_i$

---

---

**Algorithm 2** Language Model as Visual Explainer (LVX)-Test

---

**Input:** Vision model $f = g \cdot h$, a test sample $\mathbf{x}_{ts}$ and explanatory trees $T_i$ for each category.
**Output:** A explanatory tree $T$ for test sample.
1: Prediction on $\mathbf{x}_{ts}$: $\mathbf{q} = g(\mathbf{x}_{ts})$ and $\hat{y} = h(\mathbf{q})$.
2: Find the top-matched sub-tree in $T_{\hat{y}}$

$$T^* = \operatorname*{argmin}_{T^* \subseteq T_{\hat{y}}} \sum_{i=1}^k D(\mathbf{q}, P_{v_i})$$
$$s.t. \quad T^* = \{V^*, E^*\}, v_i \in V^*, |V^*| = k$$

3: **return** $T^*$ as the prediction explanation.

---

prediction rationales. This sample-wise attribute tree process enables the identification of pertinent attributes and explanations linked to each test sample, providing insights into the decision-making process of the model.

In summary, the LVX employs an iterative approach to construct category-level trees, leveraging the knowledge of LLMs. These trees are then utilized during the test stage to extract relevant explanations. This methodology enables us to gain understanding of the model's decision-making process by revealing the underlying visual attributes and rationales supporting its predictions.

## D    Data Collection and Analysis

### D.1    Annotation Creation

The creation of test annotations for three datasets involved a semi-automated approach implemented in two distinct steps. This process was a collaborative effort between human annotators, a language model (ChatGPT), and CLIP [53], ensuring both efficiency and reliability.

1. **Concept Tree Creation.** In this first step, we utilized ChatGPT [4] to generate initial attribute trees for each class with the category name, detailed Section 3.1.

2. **Attribute Verification.** To determine whether an attribute was present or absent in an image, we employed an ensemble of predictions from multiple CLIP [53] models [5]. We filtered the top-5 attributes predicted by CLIP and sought human judgments to verify their correctness. To streamline this process, we developed an annotation tool with a user interface, which is illustrated in Figure 7.

3. **Manual Verification.** In this phase, annotators examined the accuracy of existing attributes and introduced new relevant ones to enrich the concept trees. Subsequently, human annotators conducted a thorough review, refinement, and systematic organization of the attribute trees for each class.

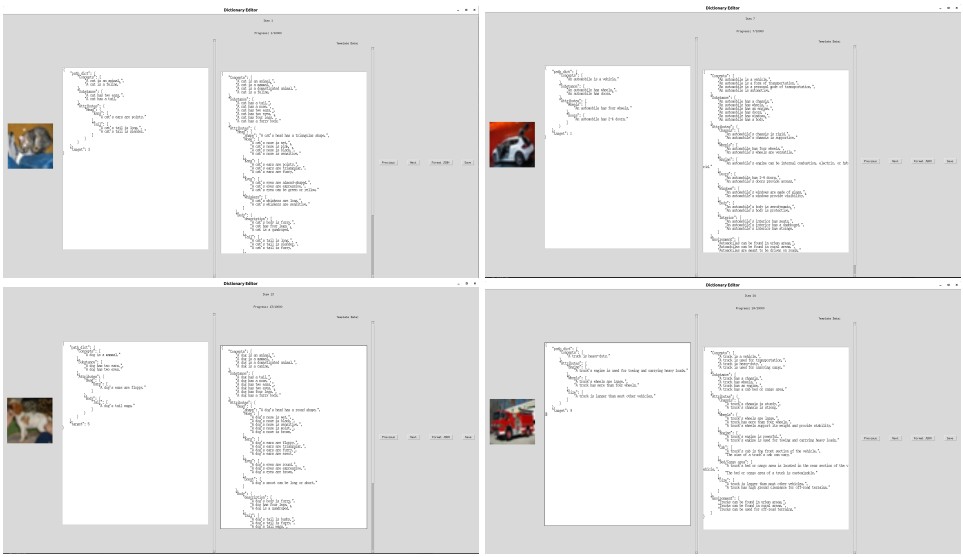

Figure 7: Software tool interface for parse tree annotation.

Table 8: Support set Dataset Statistics.

| Dataset Name | No. Categories | No. Attributes | No. Images |
|---|---|---|---|
| CIFAR-10 Support | 10 | 289 | 14,024 |
| CIFAR-100 Support | 100 | 2,359 | 19,168 |
| ImageNet Support | 1,000 | 26,928 | 142,034 |

## D.2 Support data

**Data Collection.** Our `LVX` model utilizes a custom support set for each task, created through a reject sampling method. Initially, images are generated using either Bing or the Stable Diffusion Model. Subsequently, CLIP is applied to determine if the CLIP score exceeds 0.5, based on the raw cosine similarities calculated by averaging CLIP `ViT-B/32,ViT-B/16,ViT-B/14`. If the score is above the specified threshold, the image is retained; otherwise, it is discarded.

To optimize the image collection process, we merge the retrieved images from all models, leading to time and effort savings. This approach allows us to reuse an image already in the dataset if it matches an attribute generated by the LLM. As a result, after the initial models have finished collecting data, subsequent models can simply pull relevant images from this existing pool instead of collecting new ones, saving both time and effort.

---

[4] https://chat.openai.com/
[5] https://github.com/mlfoundations/open_clip

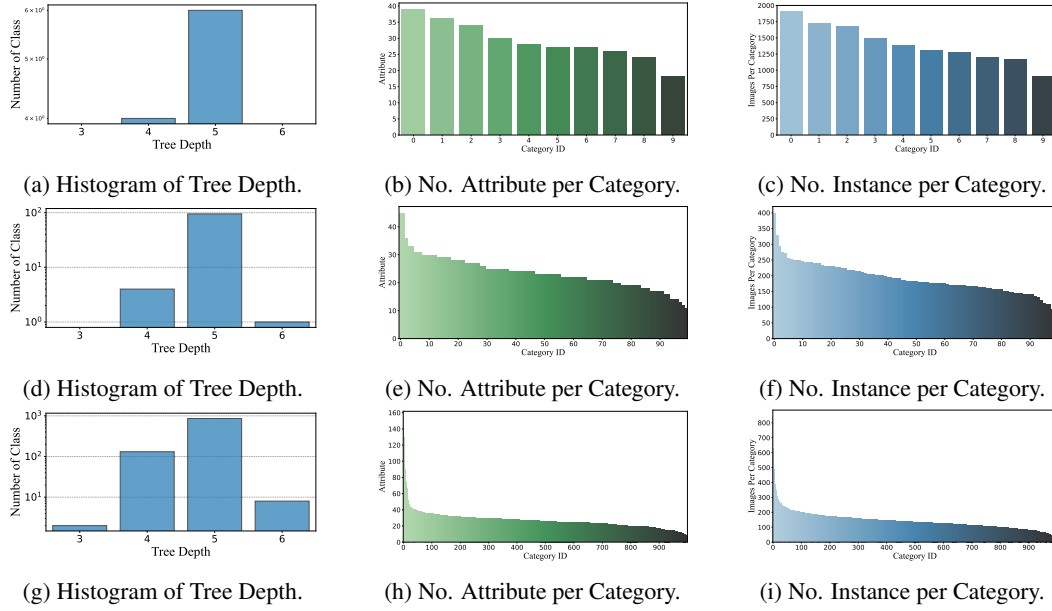

Figure 8: Statistics of the support dataset sets of (Row1) CIFAR10, (Row2) CIFAR100 and (Row3) ImageNet. We examine the (a,d,g) Tree Depth, (b,e,h) Number of Attributes for each category and (c,f,i) Number of image for each category to demonstrate the diversity of collected attributes and the completeness of hierarchical annotations.

**Data Statistics.** We present the statistics of the support datasets collected for CIFAR10, CIFAR100, and ImageNet, highlighting the diversity and comprehensiveness of our dataset. Table 8 and Figure 8 provide an overview of these statistics. Specifically, we include the number of attributes, the number of samples for each category, the total number of samples in the dataset, as well as the distribution of tree depths. This rich collection of data showcases the diverse range of attributes and categories covered in our dataset, making it a valuable resource for training and evaluation purposes.

### D.3 Limitations of Support Dataset Collection

While collecting a newly curated dataset can be advantageous for our tasks, it is important to acknowledge certain limitations when using such datasets for explanation purposes. Three key limitations arise: false positive simages, the presence of out-of-distribution samples and potential bias in the dataset.

- **False Positive Images.** We observed that both Bing and the Stable Diffusion model occasionally generate imperfect images from textual descriptions, manifesting incorrect or entangled patterns. For instance, the word "crane" could represent either a construction machine or a bird, leading to ambiguity. Furthermore, an image described as "a dog with a long tail" could potentially include not only the tail but also the head and legs, reflecting a broader scope than intended.

- **Out-of-Distribution Samples.** Newly collected datasets may include samples that are out-of-distribution, i.e., they do not align with the source data distribution of interest. These out-of-distribution samples can introduce challenges in generating accurate and reliable explanations. As a result, explanations based solely on a newly collected dataset may not generalize well to unseen instances outside the support dataset distribution.

- **Data Bias.** Biases in the collection process or the underlying data sources can inadvertently influence the dataset, leading to biased explanations. Biases emerge due to various factors, such as data collection source, or imbalances in attribute distributions. Consequently, relying solely on a newly collected dataset for explanations may introduce unintended biases into the interpretation process.

**Solutions.** To deal with mistakes in the gathered images, we used two approaches. First, we used the CLIP model to sift through the images because it's good at understanding how close an image is to text concepts, helping us remove most mixed-up and incorrect images. Second, we manually sorted out words that have more than one meaning. For instance, we made it clear whether "crane" refers to the bird or the machine by labeling it as "crane (bird)" or "crane (machine)".

To mitigate the challenges posed by OOD samples and data bias, we adopt a cautious approach. Specifically, we do not directly train our models on the newly collected support dataset. Instead, we utilize this dataset solely for the purpose of providing disentangled attributes for explanations. By decoupling the training data from the support data, we aim to reduce the impact of OOD samples and potential data biases, thus promoting a more robust and unbiased analysis.

Despite these limitations, our emphasis on obtaining a dataset with distinct attributes sharpens our analysis and interpretation of model behavior. This approach allows us to extract meaningful insights in a controlled and clear way.

# E   User Study

The evaluation of visual decision-making and categorization can be uncertain and subjective, posing challenges in assessing the quality of explanations. To address this, we conducted a user study to verify the plausibility of our explanations.

**Experiment Setup.** Our study compared the performance of LVX, with three others: `Subtree`, `TrDec`, and a new baseline called `Single`. The key difference with the `Single` baseline is that it utilizes only the nearest neighbor node from the parse tree for its output. This contrasts with LVX, which employs the top-k neighbor nodes from the parse tree.

We recruited 37 participants for this study. Each was asked to respond to 15 questions, each with one image and 4 choices. In each question, they choose the explanation that best matched the image, based on their personal judgment. The format for each choice was arranged as "`The image is a <CLASS_NAME> because <EXPLANATION>.`".

**Results.** The user study results, as shown in Table 9, clearly indicate the superiority of the LVX method. It was selected by participants 57.66% of the time, a significantly higher rate compared to the other methods included in the study.

Table 9: User Study Results.

| Method | Choice Percentage |
|---------|-------------------|
| Subtree | 3.78% |
| TrDec | 22.88% |
| Single | 15.68% |
| LVX | **57.66%** |

# F   Experiments on Out-of-Distribution (OOD) Evaluation

In this section, we evaluate our calibrated model's performance in Out-of-Distribution (OOD) scenarios, focusing on its robustness and ability to generalize. This evaluation is conducted using a ResNet50 and ViT-S trained on ImageNet, with and without calibration training, and tested on the ImageNet-A and ImageNet-Sketch datasets.

**Results.** The results of OOD generalization, quantified by Top-1 Accuracy, are listed in Table 10. For both ResNet-50 and ViT-S 16 models, we notice significant improvements in accuracy in ImageNet-A and ImageNet-S compared to the baselines. This enhancement in Out-of-Distribution generalization confirms the effectiveness of model calibration in not only improving in-domain performance (shown in Figure 7 of the main paper) but also in boosting adaptability and robustness to out-of-domain data.

Table 10: OOD Generalization Results with and without calibration by Top-1 Accuracy.

| Model | ImageNet (In-Domain) | ImageNet-A | ImageNet-S |
|---|---|---|---|
| | Baseline/**Calibrated** | Baseline/**Calibrated** | Baseline/**Calibrated** |
| ResNet-50 | 76.13/**76.54**(+0.41) | 18.96/**23.32**(+4.36) | 24.10/**31.42**(+7.32) |
| ViT-S 16 | 77.85/**78.21**(+0.36) | 13.39/**18.72**(+5.33) | 32.40/**37.21**(+4.81) |

# G   Experiments on Chest X-Ray Diagnosis

In this section, we evaluate the performance of our LVX method in security-critical domains, specifically medical image analysis. We train neural networks for chest X-ray diagnosis and utilize LVX to interpret and calibrate the predictions.

We adopted the DenseNet-121 architecture for disease diagnosis in our study. The model was trained on the Chestx-ray14 dataset [78], which consists of chest X-ray images encompassing 14 diseases, along with an additional "No Finding" class. The DenseNet-121 architecture is specifically designed to generate 14 output logits corresponding to the different diseases. During training, we employed a weighted binary cross-entropy loss [78] for each disease category to optimize the model.

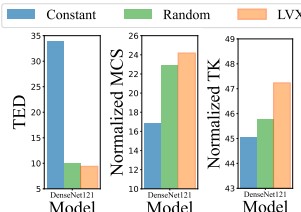

Figure 9: Explanation performance comparison on Chestx-ray14 dataset.

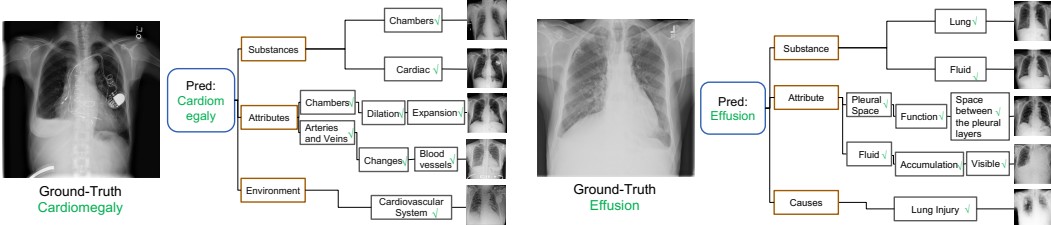

Figure 10: Explanation examples for the chest xray diagnosis task.

For optimization, we utilized the Adam optimizer [36] with an initial learning rate of 1e-4, a weight decay of 1e-5, and a batch size of 32. The model underwent training for a total of 18 epochs.

**Model Explanation.** To enhance interpretability, we incorporated our LVX framework into the model. Instead of acquiring images from online sources, we gathered the support set directly from the training data. To accomplish this, we utilized a parse tree generated by the ChatGPT language model. Leveraging this parse tree, we applied a MedCLIP [80] model to retrieve the most relevant images for each attribute from the training set. These retrieved images served as our support sets for the LVX framework.

Compared to applying the LVX framework on single-label classification, the Chestx-ray14 dataset poses a multi-label classification challenge. In this dataset, each sample can belong to multiple disease categories simultaneously. Therefore, we modified the LVX framework to accommodate and handle the multi-label nature of the classification task.

Specifically, for each input image $\mathbf{x}$, we predict its label $\hat{y} = f(\mathbf{x}) \in \{0, 1\}^{14}$. To create the visual parse tree, we begin by establishing the root node. If all elements of $\hat{y}$ are 0, the root node is set to "No Findings". Conversely, if any element of $\hat{y}$ is non-zero, the root node is labeled as "has Findings". For each positive finding, we construct a separate parse tree, with these sub-trees

becoming the children nodes of the root. By combining these sub-trees, we obtain a comprehensive and coherent explanation for the image. This modification enables us to effectively handle the multi-label nature of the classification task, providing meaningful and interpretable explanations for images with multiple positive findings.

To establish the ground-truth explanation label, we adopt a MedCLIP [80] model to filter the top-5 attributes for each positive finding of the image. These attributes are then organized into a tree structure. This approach serves as an automatic explanation ground-truth, thereby eliminating the requirement for manual annotations from domain experts.

In addition to providing explanations, we aim to calibrate the model predictions with the parse tree. To achieve this, we apply a modified hierarchical contrastive loss individually on each finding. We then calculate the average of these losses, which serves as our overall loss term. We thus fine-tune the model for 3 epochs using the hieracical term and weighted cross-entropy.

**Explanation Results.** We compare the explanation performance of our proposed LVX against the Random and Constant baselines. The numerical results, depicted in Figure 9, highlight the superiority of our LVX approach.

Additionally, we showcase the parsed visual tree in Figure 10, to provide a clearer illustration of our results. Notably, our approach effectively interprets the decision-making process of black-box neural networks. For instance, in the case on the right, our method accurately identifies the presence of *visible fluid* in the lung space and establishes its relevance to the model's prediction. Consequently, LVX enables clinical professionals to make well-informed justifications for their patients, enhancing the overall decision-making process.

**Calibration Results.** Table 11 presents the comparison between the baseline models and the model with calibration, measured in terms of the Area Under the Curve (AUC) score for each disease type. The AUC score provides a measure of the model's ability to discriminate between positive and negative cases.

The calibrated model shows notable improvements in several disease types compared to the baseline. Notably, Hernia demonstrates the most significant improvement, with an AUC score of 0.936 compared to 0.914 for the baseline. This indicates that the calibration process has enhanced the model's ability to accurately detect Hernia cases.

In summary, the LVX method markedly improves model accuracy, demonstrated by enhanced calibration performance across different disease types. The integration of visual attributes boosts both accuracy and reliability of predictions, leading to better diagnostic results. These findings underscore the LVX method's potential to elevate model performance, particularly in medical diagnostics.

Table 11: Model performance with and without calibration. AUC scores are reported for each disease type. *Avg.* indicates the average score.

| Finding | Baseline | LVX |
|---|---|---|
| Atelectasis | 0.767 | **0.779** |
| Consolidation | 0.747 | **0.755** |
| Infiltration | 0.683 | **0.698** |
| Pneumothorax | 0.865 | **0.873** |
| Edema | 0.845 | **0.851** |
| Emphysema | 0.919 | **0.930** |
| Fibrosis | 0.832 | **0.830** |
| Effusion | 0.826 | **0.831** |
| Pneumonia | 0.721 | **0.719** |
| Pleural Thickening | 0.784 | **0.793** |
| Cardiomegaly | 0.890 | **0.894** |
| Nodule | 0.758 | **0.776** |
| Mass | 0.814 | **0.830** |
| Hernia | 0.914 | **0.936** |
| Avg. | 0.812 | **0.821** |

# H   Experiments on Self-supervised models

In this section, we focus on self-supervised models to assess their interpretability. Differing from supervised models, self-supervised models develop representations without labeled data. Our aim is to understand the interpretability of these representations and uncover the underlying structures derived from the input data alone.

**Model to be Explained.** Our objective is to offer a comprehensive explanation for self-supervised models trained on ImageNet-1k. These models include ResNet50 trained using SimCLR [11], BYOL [24], SwAV [8], MoCov3 [12], DINO [9], and ViT-S trained with MoCov3 [13] and DINO [9]. The networks are subsequently fine-tuned through linear probing while keeping the *backbone fixed*. We then utilize our LVX to provide explanations for their predictions. Additionally, we compare these self-supervised models with their supervised counterparts to highlight the differences in representation between the two approaches.

**Numerical Results.** Table 12 presents the results of self-supervised models. Our analysis reveals a strong correlation between the explanatory performance and the overall model accuracy.

However, we also noticed that self-supervised models based on transformer architecture exhibit greater attribute disentanglement compared to supervised models, despite potentially having slightly lower performance. This phenomenon is evident when comparing the DINO ViT-S/16 model and the supervised ViT-S/16 model within the context of tree parsing explanation. Although the DINO ViT-S/16 model shows slightly lower overall performance, it outperforms the supervised model in terms of providing accurate attribute explanation.

These results underscore the potential benefits of self-supervised learning in uncovering meaningful visual attributes without explicit supervision. While self-supervised models may exhibit marginally lower performance on certain tasks, their ability to capture rich visual representations and attribute disentanglement highlights their value in understanding complex visual data.

Table 12: Explanation performance analysis of self-supervised models utilizing linear probing.

| Method | Top-1 Acc | TED↓ | MCS↑ | TK ↑ |
|---|---|---|---|---|
| ResNet50-SimCLR | 69.2 | 9.38 | 23.72 | 46.53 |
| ResNet50-BYOL | 71.8 | 9.29 | 24.66 | 48.29 |
| ResNet50-MoCov3 | 74.6 | 9.19 | 25.59 | 50.17 |
| ResNet50-DINO | 75.3 | 9.14 | 25.77 | 50.71 |
| ResNet50-SwAV | 75.3 | 9.15 | 25.82 | 50.69 |
| ResNet50-Sup | 76.1 | 9.09 | 25.99 | 51.19 |
| ViT-S/16-MoCov3 | 73.2 | 9.16 | 25.25 | 49.40 |
| ViT-S/16-DINO | 77.0 | 8.99 | 26.61 | 52.05 |
| ViT-S/8-DINO | 79.7 | 8.89 | 27.62 | 53.95 |
| ViT-S/16-Sup | 77.9 | 9.10 | 25.73 | 50.34 |

# I   Raw Results

This section presents the raw numerical results for Figure 5, as depicted in the main paper. Specifically, Table 13 provides the results for CIFAR-10, Table 14 for CIFAR-100, and Table 15 for ImageNet. We also observed that larger networks within the same model family deliver better results. As the models improve, so does the accuracy of the explanations, suggesting that larger networks facilitate more effective explanations. This is demonstrated by the increase in MCS and TK scores as ResNet deepens on CIFAR-100 and ImageNet, aligning with the general belief that larger neural networks offer enhanced generalization and structural representation capabilities.

# J   Experimental Setup

In this section, we provide detailed information about our experimental setup to ensure the reproducibility of our method.

Table 13: Explanation performance comparison on CIFAR-10.

| Model | TED↓ | | | MCS↑ | | | Tree Kernel↑ | | |
|---|---|---|---|---|---|---|---|---|---|
| | rand. | const. | LVX | rand. | const. | LVX | rand. | const. | LVX |
| VGG13 | 9.21 | 32.97 | 8.21 | 28.98 | 18.77 | 32.31 | 59.49 | 58.41 | 63.43 |
| VGG16 | 9.23 | 32.89 | 8.14 | 29.11 | 19.11 | 32.55 | 59.34 | 59.55 | 63.79 |
| VGG19 | 9.15 | 32.85 | 8.15 | 30.67 | 19.10 | 31.78 | 59.44 | 59.39 | 63.32 |
| ResNet18 | 9.21 | 32.90 | 8.52 | 28.95 | 18.92 | 30.24 | 58.94 | 58.87 | 61.19 |
| ResNet34 | 9.21 | 32.92 | 8.16 | 28.95 | 18.98 | 32.07 | 59.06 | 59.01 | 63.27 |
| ResNet50 | 9.21 | 32.89 | 8.44 | 28.96 | 19.00 | 31.09 | 59.16 | 59.21 | 62.06 |
| DenseNet121 | 9.19 | 32.89 | 8.20 | 29.09 | 19.11 | 32.13 | 59.53 | 59.48 | 63.44 |
| DenseNet161 | 9.20 | 32.88 | 8.19 | 29.07 | 19.11 | 32.12 | 59.35 | 59.48 | 63.73 |
| DenseNet169 | 9.21 | 32.88 | 8.18 | 29.25 | 19.08 | 32.13 | 59.52 | 59.46 | 63.46 |
| MobileNet_v2 | 9.20 | 32.89 | 8.41 | 29.24 | 19.09 | 31.61 | 59.53 | 59.38 | 61.87 |
| GoogLeNet | 9.23 | 32.96 | 8.41 | 28.66 | 18.86 | 30.75 | 58.62 | 58.71 | 61.38 |
| Inception_v3 | 9.20 | 32.89 | 8.39 | 29.19 | 19.03 | 31.02 | 59.37 | 59.27 | 61.85 |

Table 14: Explanation performance comparison on CIFAR-100.

| Model | TED↓ | | | MCS↑ | | | Tree Kernel↑ | | |
|---|---|---|---|---|---|---|---|---|---|
| | rand. | const. | LVX | rand. | const. | LVX | rand. | const. | LVX |
| ResNet20 | 9.61 | 28.77 | 8.96 | 22.65 | 17.60 | 24.89 | 45.52 | 46.96 | 47.70 |
| ResNet32 | 9.57 | 28.67 | 8.86 | 22.84 | 17.92 | 25.39 | 46.45 | 47.87 | 48.58 |
| ResNet44 | 9.54 | 28.60 | 8.81 | 23.48 | 18.34 | 25.97 | 47.42 | 48.87 | 49.79 |
| ResNet56 | 9.52 | 28.54 | 8.83 | 23.87 | 18.60 | 26.47 | 48.04 | 49.58 | 50.17 |
| MBNv2-x0.5 | 9.55 | 28.58 | 8.87 | 23.43 | 18.19 | 25.50 | 47.13 | 48.58 | 49.08 |
| MBNv2-x0.75 | 9.43 | 28.43 | 8.76 | 24.47 | 18.87 | 26.71 | 49.19 | 50.55 | 51.21 |
| MBNv2-x1.0 | 9.48 | 28.35 | 8.73 | 24.35 | 19.02 | 27.09 | 49.27 | 50.68 | 51.47 |
| MBNv2-x1.4 | 9.43 | 28.16 | 8.65 | 24.87 | 19.47 | 27.41 | 50.52 | 52.08 | 52.91 |
| RepVGG A0 | 9.44 | 28.28 | 8.74 | 24.65 | 19.21 | 26.84 | 49.78 | 51.37 | 52.01 |
| RepVGG A1 | 9.42 | 28.21 | 8.72 | 25.27 | 19.59 | 27.43 | 50.63 | 52.17 | 52.81 |
| RepVGG A2 | 9.40 | 28.08 | 8.70 | 25.46 | 19.82 | 27.99 | 51.26 | 52.87 | 53.04 |

## J.1 Evaluation Metrics

To evaluate the effectiveness of our proposed tree parsing task, we have developed three metrics that leverage conventional tree similarity and distance measurement techniques.

- **Tree Kernels (TK)**: Tree Kernels (TK) evaluate tree similarity by leveraging shared sub-structures, assigning higher scores to trees with common subtrees or substructures. To enhance the match, we set the decaying factor for two adjacent tree layers to 0.5, where larger values lead to better matches. Let's define the subtree kernel mathematically:

Table 15: Explanation performance comparison on ImageNet.

| Model | TED↓ | | | MCS↑ | | | Tree Kernel↑ | | |
|---|---|---|---|---|---|---|---|---|---|
| | rand. | const. | LVX | rand. | const. | LVX | rand. | const. | LVX |
| ResNet18 | 9.83 | 34.15 | 9.30 | 22.52 | 16.82 | 23.87 | 45.32 | 44.85 | 46.85 |
| ResNet34 | 9.75 | 33.78 | 9.17 | 23.74 | 17.71 | 25.09 | 47.66 | 47.16 | 49.24 |
| ResNet50 | 9.68 | 33.58 | 9.09 | 24.59 | 18.35 | 25.99 | 49.38 | 48.97 | 51.19 |
| ResNet101 | 9.64 | 33.48 | 9.04 | 24.94 | 18.66 | 26.51 | 50.25 | 49.77 | 51.99 |
| ViT-T16 | 10.42 | 35.44 | 9.99 | 15.07 | 11.25 | 15.91 | 30.30 | 29.92 | 31.24 |
| ViT-S16 | 9.69 | 33.61 | 9.10 | 24.16 | 18.01 | 25.73 | 48.53 | 48.05 | 50.34 |
| ViT-B16 | 9.62 | 33.37 | 8.99 | 25.20 | 18.79 | 27.01 | 50.64 | 50.22 | 52.76 |
| ViT-L16 | 9.45 | 32.84 | 8.79 | 27.27 | 20.35 | 29.29 | 54.83 | 54.36 | 57.14 |

---

**Tree Kernel:** Given two trees, $T_1$ and $T_2$, represented as rooted, labeled, and ordered trees, we define the subtree kernel as follows:

$$TK(T_1, T_2) = \sum_T \sum_{T'} \theta(T, T') \times \theta(T, T') \times \lambda^{\max(\mathrm{depth}(r), \mathrm{depth}(r'))}$$

Let $TK(T_1, T_2)$ denote the similarity between subtrees $T_1$ and $T_2$ using the subtree kernel. Additionally, let $\theta(T, T')$ represent the count of shared common subtrees between trees $T$ and $T'$. Furthermore, let $r$ and $r'$ be the roots of $T$ and $T'$ respectively. $\lambda < 1.0$ is the decaying factor that make sure the tree closer to the root hold greater significance.

The $\theta(T, T')$ is computed recursively as follows:

1. If both $T$ and $T'$ are leaf nodes, then $\theta(T, T') = 1$ if the labels of $T$ and $T'$ are the same, and 0 otherwise.
2. If either $T$ or $T'$ is a leaf node, then $\theta(T, T') = 0$.
3. Otherwise, let $\{T_1, T_2, \ldots, T_n\}$ be the child subtrees of $T$, and $\{T'_1, T'_2, \ldots, T'_{n'}\}$ be the child subtrees of $T'$.
   - If the labels of $r$ and $r'$ are the same, then $\theta(T, T')$ is the sum of the products of $\theta(T_i, T'_j)$ for all combinations of $i$ and $j$, where $i$ ranges from 1 to $n$ and $j$ ranges from 1 to $n'$.
   - If the labels of $r$ and $r'$ are different, then $\theta(T, T')$ is 0.
   - Additionally, if $T$ and $T'$ are isomorphic (have the same structure), then $\theta(T, T')$ is incremented by 1.

In the paper, the Tree Kernel (TK) score is normalized to accommodate trees of different sizes. The normalized TK score is computed as: $\frac{TK(T_{pred}, T_{gt}) \times 100}{\sqrt{TK(T_{pred}, T_{pred}) TK(T_{gt}, T_{gt})}}$. The kernel value serves as a measure of similarity, where higher values indicate greater similarity.

- **Maximum Common Subgraph (MCS)**[54, 33]: The Maximum Common Subgraph (MCS) identifies the largest shared subgraph between two trees, measuring the similarity and overlap of their hierarchical structures. Here's the mathematical definition of the Maximum Common Subgraph:

> **Maximum Common Subgraph:** Given two trees, $T_1 = (V_1, E_1)$ and $T_2 = (V_2, E_2)$, where $V_1$ and $V_2$ are the sets of vertices and $E_1$ and $E_2$ are the sets of edges for each graph, respectively, we define the **Maximum Common Subgraph** as:
>
> $$\text{MCS}(T_1, T_2) = (V_{\text{MCS}}, E_{\text{MCS}})$$
>
> maximize $\quad |V_{\text{MCS}}|$
>
> subject to $\quad V_{\text{MCS}} \subseteq V1 \quad$ and $\quad V_{\text{MCS}} \subseteq V2$
>
> $\qquad\qquad\quad E_{\text{MCS}} \subseteq E1 \quad$ and $\quad E_{\text{MCS}} \subseteq E2$
>
> $\qquad\qquad\quad$ For any pair of vertices $u, v$ in $V_{\text{MCS}}$, if $(u, v)$ is an edge in $E_{\text{MCS}}$,
>
> $\qquad\qquad\qquad$ then $(u, v)$ is an edge in $T_1$ and $T_2$, and vice versa.

In our paper, we report the normalized MCS score as our measurement of tree similarity $\frac{|\text{MCS}(T_{pred}, T_{gt})| \times 100}{\sqrt{|T_{pred}||T_{gt}|}}$, where a higher score indicates greater similarity between the graphs. Here, $|\cdot|$ represents the number of nodes in a tree. We employ this normalization to address the scenario where one tree is significantly larger and encompasses all other trees as subtrees. By dividing the MCS score by the square root of the product of the numbers of nodes in the predicted tree ($T_{\text{pred}}$) and the ground truth tree ($T_{\text{gt}}$), we ensure a fair comparison across trees of varying sizes.

- **Tree Edit Distance (TED)** [6]: The Tree Edit Distance (TED) quantifies the minimum number of editing operations required to transform one hierarchical tree into another. It measures the structural dissimilarity between trees by considering node and edge modifications, insertions, and deletions. With smaller TED, the two graphs are more similar. Let's define the Tree Edit Distance formally:

> **Tree Edit Distance:** Given two trees $T_1$ and $T_2$, represented as rooted, labeled, and ordered trees, we define the Tree Edit Distance between them as $TED(T_1, T_2)$.
>
> The $TED(T_1, T_2)$ is computed recursively as follows:
>
> - If both $T_1$ and $T_2$ are empty trees, i.e., they do not have any nodes, then $TED(T_1, T_2) = 0$.
> - If either $T_1$ or $T_2$ is an empty tree, i.e., it does not have any nodes, then $TED(T_1, T_2)$ is the number of nodes in the non-empty tree.
> - Otherwise, let $r_1$ and $r_2$ be the roots of $T_1$ and $T_2$, respectively. Let $T_1^{'}$ and $T_2^{'}$ be the subtrees obtained by removing the roots $r_1$ and $r_2$ from $T_1$ and $T_2$, respectively.
>   * If $r_1 = r_2$, then $TED(T_1, T_2)$ is the minimum among the following values:
>     · $TED(T_1^{'}, T_2^{'})$ + $TED$(children of $r_1$, children of $r_2$), where $TED$(children of $r_1$, children of $r_2$) is the TED computed recursively between the children of $r_1$ and $r_2$.
>     · $1 + TED(T_1^{'}, T_2^{'})$, which represents the cost of deleting the root $r_1$ and recursively computing TED between $T_1^{'}$ and $T_2^{'}$.
>     · $1 + TED(T1, T_2^{'})$, which represents the cost of inserting the root $r_2$ into $T_1$ and recursively computing TED between $T_1$ and $T_2^{'}$.
>     · $1 + TED(T_1^{'}, T_2)$, which represents the cost of deleting the root $r_1$ and inserting the root $r_2$ into $T_2$, and recursively computing TED between $T_1^{'}$ and $T_2$.
>   * If $r_1 \neq r_2$, then $TED(T_1, T_2)$ is the minimum among the following values:
>     · $TED(T_1^{'}, T_2^{'})$ + $TED$(children of $r_1$, children of $r_2$), where $TED$(children of $r_1$, children of $r_2$) is the TED computed recursively between the children of $r_1$ and $r_2$.
>     · $1 + TED(T_1^{'}, T_2)$, which represents the cost of deleting the root $r_1$ and recursively computing TED between $T_1^{'}$ and $T_2$.
>     · $1 + TED(T_1, T_2^{'})$, which represents the cost of inserting the root $r_2$ into $T_1$ and recursively computing TED between $T_1$ and $T_2^{'}$.

The $TED(T_1, T_2)$ is the final result obtained after applying the above recursive computation.

## J.2 Model Checkpoints

For our experiments, we utilize publicly available pre-trained models. Specifically, we employ CIFAR10 models from `https://github.com/huyvnphan/PyTorch_CIFAR10`, CIFAR100 models from `https://github.com/chenyaofo/pytorch-cifar-models`, and ImageNet models from `torchvision` package and `timm` package. The self-supervised models are downloaded from their respective official repositories.

## J.3 Explanation Baselines

To explain visual models using tree-structured language without annotations, we devised four basic baselines, `Random`, `Constant`, `Subtree` and `TrDec`, for comparison with our LVX method.

**Random Baseline.** The Random baseline generates random explanations by predicting an image's category and randomly sampling 5 nodes from its category-specific tree. The connected nodes form a tree-structured explanation, providing a performance baseline for random guessing.

**Constant Baseline.** The Constant baseline Produces a fixed tree-structured clue for images of the same class, using an initial explanatory tree $T_i^{(0)}$ as a template. This baseline assesses LVX against a non-adaptive, static approach.

**Subtree Baseline.** This method involves selecting the most common subtree from the test set for explanations, testing the efficacy of using frequent dataset patterns for generic explanations.

**TreDec Baseline.** Based on [79], the `TrDec` strategy implements a tree-topology RNN decoder over an image encoder. In the absence of hierarchical annotations, this baseline uses the CLIP model to verify nodes in the template trees, which act as pseudo-labels for training. This method focuses on the effectiveness of a structured decoding process in explanation generation.

This comparison demonstrates the effectiveness of `LVX` in creating explanations tailored to individual image content, clearly outperforming methods based on random guessing, static templates, and basic learning-based approaches.

## K   Limitations

Our system, LVX, depends on an external Large Language Model (LLM) to provide textual explanations. While this integration adds significant functionality, it also introduces the risk of inaccuracies. The LLM may not always deliver correct information, leading to potential misinformation or erroneous explanations.

Additionally, our approach involves generating explanations based on the last embedding layer of the neural network. This method overlooks the comprehensive, multi-level hierarchical structure of deep features, potentially simplifying or omitting important contextual data that could enhance the understanding of the network's decisions.

