# OpenReview forum: "Language Model as Visual Explainer"
_NeurIPS.cc/2024/Conference — NeurIPS 2024 poster_

### Official Review · Reviewer_FVKA · 2024-07-03

**Soundness:** 3
**Presentation:** 2
**Contribution:** 3
**Rating:** 5
**Confidence:** 4

**Summary:**

This paper builds an attribute tree using LLMs to explain image classifiers. To this end, the paper designs a framework that expands and prunes nodes for the tree. Using this framework, the paper provides a richly annotated version of CIFAR and ImageNet, which is 5 times more expressive than prior works. With this attribute tree, the paper outperforms previous decision tree approaches such as NBDT.

**Strengths:**

- New datasets with rich attributes could be valuable for the community, though the necessity of such large attribute sets should be further justified (W3).
- Using rich attributes, the paper outperforms previous NN-based decision trees, such as NBDT.
- The paper is quite dense, containing a lot of information.

**Weaknesses:**

1. Explanation is more than an attribute tree

The paper focuses on building attribute trees using LLMs. However, there are more diverse ways to interpret the behaviors of vision models, particularly using language. For example, [1,2] use vision-language models to understand the failures of vision models through language explanations. [3-5] use language to derive concepts for concept bottleneck models (CBMs), though this paper only cites [5]. [6] generates language to explain the neurons in the network. Among these approaches, the paper should discuss the advantages and drawbacks of the proposed method of creating an attribute tree.

For this reason, I believe the current paper title is overstated. I recommend clarifying its scope in the title to focus on the construction of hierarchical attributes and decision trees.

[1] Kim et al. Discovering and Mitigating Visual Biases through Keyword Explanation. CVPR 2024.\
[2] Wiles et al. Discovering Bugs in Vision Models using Off-the-shelf Image Generation and Captioning. arXiv 2023.\
[3] Yuksekgonul et al. Post-hoc Concept Bottleneck Models. ICLR 2023.\
[4] Oikarinen et al. Label-Free Concept Bottleneck Models. ICLR 2023.\
[5] Yang et al. Language in a Bottle: Language Model Guided Concept Bottlenecks for Interpretable Image Classification. CVPR 2023.\
[6] Hernandez et al. Natural Language Descriptions of Deep Visual Features. ICLR 2022.


---
2. Insufficient qualitative insights

The paper mostly focuses on quantitative metrics such as plausibility, faithfulness, and stability, which support the validity of the proposed explanation. However, an explanation is only meaningful when it provides new insights to humans. In this sense, the paper should carefully examine individual data points and their corresponding explanations and discuss the lessons learned from them.

I know that Figure 6 provides some examples of decision trees. However, the paper could delve deeper, for example, by addressing: 1) What are the common issues with current classifiers? 2) What are the differences between different models, such as ResNet vs. ViT? As the paper has built a tool, there are numerous directions to explore using it.

---
3. Necessity of enormous attributes

The paper creates datasets with rich attributes (H-XX), which are 5 times more expressive than the prior datasets (DR-XX). However, do we really need all these attributes? More analysis and justification of the attributes should be provided, clearly showing the limitations of prior works.

**Questions:**

Can the method be improved using multimodal models like GPT-4V instead of language-only models like GPT-4?

**Limitations:**

Discussed in the Appendix.

---

> ### Author Rebuttal · Authors · 2024-08-05
>
> We truly appreciate the suggestion and acknowledgment from R-FVKA regarding the dataset and method.
>
> `>>> Q1` **Explanation beyond Attribute Tree**
>
> `>>> A1` Amazing question. We appreciate the valuable papers mentioned by the reviewer and will cite them in our final version. While all [1-6] use natural language to explain visual models, our method has three key distinctions:
>
> 1. **No Training Required:** Compared to [3-5], LVX eliminates the need for human annotations and additional model training. It works as a plug-and-play solution.
> 2. **Use of Large Language Model:** Compared with [1-4, 6], we use large  language models as automatic tools to explain vision models, providing rich knowledge and  open-vocabulary capabilities.
> 3. **Hirachical structured explanation:** While [1-6] provide single-round, single-grain explanations, LVX offers hierarchical, multi-grained explanations.
>
> We will cite these works properly and include a discussion in our final version.
>
> `>>> Q2` **Paper Title**
>
> `>>> A2` We truly appreciate the suggestion. We will revise our title as `Language Model as Hierachical Visual Concept Explainer` to better reflect the scope of our work.
>
> `>>> Q3` **Experiments and Insights**
>
> `>>> A3` This is a nice point. As suggested, we do two experiments to use LVX to study the common problems in current classifiers, and compare classifier from different families.
>
> **1) Common Issue**
>
> Per the advice, we identify the most frequently correct and misclassified attributes in LVX on ImageNet across 8 networks.
> - **Setup:** We present word clouds of these correct classifications and errors in `Fig 12` of rebuttal PDF.
> - **Results: Shape-bias**. Our findings indicate that current deep networks are better at recognizing `Attributes` like **color**, but they struggle with accurately recognizing object `Substances` based on their **shape** and **size**.
>
> **2) Comparing CNN and ViT**
>
> As suggested, we conducted experiments to compare CNN and Transformer models and identify which concepts they miss.
> - **Setup:** We examined 26,928 ImageNet concepts, categorized into `Concepts, Substances, Attributes, and Environments` defined in the template. We measured (1) errors in each sub-category and (2) accuracy at different tree depths $l$. Note that when $l=1$, the accuracy is the typical ImageNet top-1 accuracy. We selected DeiT-B and ConvNeXt-T because they have similar top-1 ImageNet accuracy.
>
> - **Result:** While both models show similar overall accuracy, we found that CNNs are better at recognizing local patterns, like detailed `Attributes` and `Substances` of objects. In contrast, Transformers like DeiT-B perform better on `Environments`, focusing on broader contexts.
>
> |Model|Accuracy(%)||||
> |-|-|-|-|-|
> |**Subset**|*Concepts*|*Substances*|*Attributes*|*Environments*|
> |ConvNeXt-T|*23.1* |*18.9* |**45.3**|18.1|
> |DeiT-B|22.0|15.6|35.7|**26.3**|
>
> Additionally, Transformers are more accurate at shallow depths, while CNNs excel at deeper depths. This finding aligns with earlier research showing that **CNN are biased towards textures over shape**[A].
>
> |Model|Accuracy(%)||||
> |-|-|-|-|-|
> |**Depth**| 1(ImageNet Acc)|2|3|4|5|
> |ConvNeXt-T |82.1|65.1| 46.2|**35.7**|**25.8**|
> |DeiT-B |81.8|**70.1**|**48.0**|32.7|13.9|
>
> We will incorporate them into the final version.
>
> [A] ImageNet-trained CNNs are biased towards texture; increasing shape bias improves accuracy and robustness, ICLR 2019
>
> `>>> Q4` **Use of fine-grained Attributes**
>
> `>>> A4` Thanks. Yes, collecting more fine-grained attributes are indeed useful.
>
> **1) Dataset Analysis and Comparison**
>
> We've provided more dataset statistics in `Fig 8` of Appendix.  As suggested, we also compared attributes per class between H-ImageNet and DR-ImageNet in `Fig 11` in the rebuttal PDF.
>
> This comparison clearly shows that, even the maximum number of attributes in DR-ImageNet is less than the minimum number of attributes in H-ImageNet, **max(DR-ImageNet) < min(H-ImageNet)**. It supports our dataset's greater diversity and depth.
>
> **2) Previous Limitations**
>
> Previous datasets that include attributes have limitations in two key areas:
> - **Insufficient Attribute in prior work:** DR-ImageNet, although large, has only 5,810 attributes, averaging **5.8 unique attributes per category**. In contrast, our _H-ImageNet_ has 26,928 attributes, providing a much richer dataset.
> - **Fine-grained Explanation:** Current datasets[34-37] focus on single-level, coarse grained attributes. Humans, however, provide explanations in a multi-dimensional and hierarchical manner. Our _H-ImageNet_ reflects this complexity, making it more helpful for understanding fine details.
>
>  In this scope, our _H-X_ dataset addresses these gaps by providing a richer set of attributes for more detailed analysis and applications.
>
> **3) More use case**
>
> The rich attribute data in  _H-ImageNet_  can also be used for other tasks, such as **tree label classification and model calibration**, expanding its applicability beyond its initial scope.
>
> `>>> Q5` **Using Multimodal Models for Explanation**
>
> `>>> A5` Thank you for the suggestion. We are currently exploring the use of Multimodal Large Language Models (MLLM) like GPT-4 for enhancing explanations. Here are a few potential directions:
> 1. **Automatic Tree Label Collection**: Instead of manual annotation, we can use MLLMs to generate tree-like labels from images. This method can efficiently create large-scale hierarchical annotations for evaluation and model calibration.
> 2. **Advanced Image Filtering:** LVX uses tools to synthesize images, which can sometimes contain errors. MLLMs can verify these images to ensure the attributes are correct.
> 3. **Model Error Detection:** MLLMs can spot errors by comparing tree explanation with image content. For example, if a model wrongly says a long-haired dog has no hair, MLLMs can check the image and find the mistake.
>
> We appreciate the suggestion and will continue to explore these possibilities.

---

> ### Comment · Reviewer_FVKA · 2024-08-10
> **Response to the Rebuttal**
>
> Thank you for the detailed response and experiments. I believe the construction of hierarchical concepts is valuable for XAI, despite concerns about its usefulness raised by other reviewers. Therefore, I stick to my original rating of borderline weak acceptance.
>
> I appreciate the decision to revise the paper title to better clarify the scope; the new title will be more scientific than the previous overly-hyped one.
>
> I also value the new experiments comparing models using their XAI method, which show that current models are biased towards color over shape and size, and that CNNs and ViTs excel in local and global pattern recognition, respectively. While these insights aren't entirely new, it's reassuring to see them align with prior work.

---

> > ### Author Response · Authors · 2024-08-10
> > **Great Appreciation to Reviewer**
> >
> > Dear Reviewer FVKA,
> >
> > Truly thank you for the nice words and the recognition of our efforts. We will incorporate all the suggestions to make our work more scientific.
> >
> > One more thing to double-check is that the author mentioned "stick to my original rating of weak acceptance". But the current rating is baseline acceptance rather than WA(6). Not sure if there is any misunderstanding from our side, any error in the system or R-FVKA just meant it.
> >
> > Anyway, your suggestions are valuable for us.
> >
> > Best!

---

### Official Review · Reviewer_kLjQ · 2024-07-07

**Soundness:** 4
**Presentation:** 4
**Contribution:** 4
**Rating:** 8
**Confidence:** 4

**Summary:**

The goal of the paper is to bridge the gap between human comprehension and AI decision. For this purpose, the authors propose a Language Model as Visual Explainer (LVX), an approach to interpret the internal workings of vision models through tree-structured linguistic explanations, without the need for additional model training.  The propose method leverages the collaboration between vision models and large language models (LLM) to generate these explanations. The LLM is used to outline hierarchical visual attributes, while a text-to-image API retrieves images that best match these textual descriptions. By mapping these texts and images to the vision model’s embedding space, they create a hierarchical visual embedding tree. This tree is dynamically adjusted by querying the LLM with language templates, allowing for the addition new attributes and removal of irrelevant concepts based on the model’s representations. Additionally, the authors propose a new benchmark and new metrics to demonstrate the plausibility, faithfulness, and stability of the newly introduced method.

**Strengths:**

Here are the paper's strengths:
- it introduces a novel exaplainability method leveraged by LLM
- it introduces a new benchmark and novel metrics for an efficient evaluation
- the paper is well documented and clearly written. The contributions and objectives are clearly stated.
- the review of the state of the art is comprehensive and covers most of the relevant works
- the experimental validation is extensive

**Weaknesses:**

Weaknesses:
- some concepts presented in the paper require more details

**Questions:**

Here are my concerns:
- Eq. 3: How is the loss function $L_{HMC}$ defined
- Section 4.1 -> Evaluation Metrics -> Plausability: How is the unnormalized TK score defined?
- How does your approach may cope with fine-grained datasets, such as Flowers or CUB-200? I assume that the robustness of the approach relies on the capability of the LLM to cope with fine-grained datasets. Do you think that your approach could handle such case or some modifications (fine-tuning) should be performed on the LLM in order to deal with this particular case. Please provide your insight with respect to this problem.

**Limitations:**

Limitations are related with the curation of the new dataset and imply the following aspects: false positive simages, the presence of out-of-distribution samples and potential bias in the dataset. This research work does not have any negative societal impact.

---

> ### Author Rebuttal · Authors · 2024-08-05
>
> Thank the reviewer for the positive feedback. We are delighted that R-kLjQ acknowledge our novel explainability method using LLMs, our new benchmark, and our evaluation metrics. Those encouraging words means a lot and inspires us to push our research further.
>
> `>>> Q1`**Hierarchical Contrastive Loss $L_{HMC}$**
>
> `>>> A1`Thanks. The Hierarchical Multi-label Contrastive Loss $L_{HMC}$ comes from [33]. It is designed to learns tree-structured label using supervised contrastive learning.
>
> - **Idea**:  $L_{HMC}$ applies contrastive learning to hierarchical labels. It adds higher weight to early layer nodes, to ensure feature compactness. Conversely, it gives less constraint on leaf node features. Fine-tuning with  `Eq 3`  integrates hierarchical constraints into the feature space of classifiers.
>
> - **Math**: Say, $L$ is the set of all label levels, and $l \in L$ is a level in the multi-label hierarchy. The loss for pairing an anchor image, indexed by $i$, with a positive image at level $l$ is defined as:
>
> 	$$
> 	L_{\text{pair}}(i, p_l) = -\log \frac{\exp(f_i \cdot f_{p_l} / \tau)}{\sum_{a \in A \setminus \{i\}} \exp(f_i \cdot f_a / \tau)}
> 	$$
>
> 	The hierarchical multi-label contrastive loss $L_{\text{HMC}}$ is defined as:
>
> 	$$
> 	L_{\text{HMC}} = \sum_{l \in L} \frac{1}{|L|} \sum_{i \in I} \frac{-\lambda_l}{|P_l(i)|} \sum_{p_l \in P_l} L_{\text{pair}}(i, p_l)
> 	$$
>
> `>>> Q2`**Definition of TK score**
>
> `>>> A2`Thanks for the question. Tree Kernel (TK) measures the difference between two trees, by computing the number of the common nodes for all pairs of sub-trees. Its definition is briefly mentioned from `Line 253`, and full definition in Appendix `Sec I.1`.
>
> `>>> Q3`**LVX on fine-grained Dataset**
>
> `>>> A3`Great question!
>
> **1) Challenges for LVX on fine-grained Dataset**
>
> While our method can manage fine-grained datasets conceptually, a naive adaptation of LVX faces challenges due to *limitations of existing tools*:
> 1.  **Language Limitations:**  Not all attributes can be clearly described using LLM. For example, in CUB-200, subtle variations in the shape and color of a bird’s beak might be difficult to articulate precisely with LLM.
> 2.  **Image Generation Challenges:**  Image generation models like Stable Diffusion often have limited control over fine-grained details. This can result in images that do not accurately reflect subtle distinctions, leading to errors.
>
> **2) Experiments when Assuming Perfect Tools**
>
> As suggested, we conducted an experiment on CUB-200 when assuming all tools are perfect.
>
> Specifically, because CUB-200 already provides a hierarchical attribute labels, we retrieved images directly from the dataset based on these attributes. In this way, we do not use LLMs or image generation models, avoiding the issues associated with these tools. Although this is not the original LVX version, **it simulates the case when we have perfect language and image generation abilities**.
>
> We used a pre-trained ViT-S on CUB-200 to generate explanations. For the generated tree, we measured its alignment with the annotated tree using MCS and TK scores. We also compared it to a baseline,*TrDec*, which was trained to decode the tree structure with a frozen encoder.
>
> | Model | CUB-200(MCS&#8593;/TK&#8593;) | |
> |--|--| --|
> | | **TrDec**| **LVX** |
> | ViT-S | 26.21/45.12 | **28.64/57.18**|
>
> As shown in the table, LVX performs well on fine-grained datasets, as long as both the LLM and image generation tools can accurately capture fine details.

---

> > ### Comment · Reviewer_kLjQ · 2024-08-12
> > **Acknowledgement of Rebuttal**
> >
> > I want to thank the authors for addressing all my concerns.

---

> > > ### Author Response · Authors · 2024-08-13
> > >
> > > Thank the reviewer again for your kind and encouraging comments!
> > >
> > > Best!

---

### Official Review · Reviewer_UiCA · 2024-07-11

**Soundness:** 2
**Presentation:** 2
**Contribution:** 2
**Rating:** 4
**Confidence:** 4

**Summary:**

The paper suggests a new approach to building a hierarchical tree of visual attributes (represented with language) that matches the decision-making mechanisms of classifiers. The approach is based on using an LLM to suggest a tree of attributes that are related to a specific class, and then refine the tree according to its matches to the training date (this is done by comparing image embeddings of per attribute synthetic generated images and the train images). The hierarchical explanations are then evaluated using three measures (defined by the authors): plausibility, faithfulness, and stability. The tree explanations are shown to be beneficial when fine-tuning the classifier to match better the found structure, which slightly improves performance. The authors also exemplified the usage of the explanation for characterizing miss classification.

**Strengths:**

* The idea of constructing a language description tree of a classifier is great! (although I'm afraid the method mainly visualizes the training images and not the network operations, see comments in weaknesses section)

* The use of LLMs to generate and refine explanations inssures the generation of human-like descriptions, wich are human understandable. I liked the idea of refining the tree using the LLM.

**Weaknesses:**

* My main concern is that unintentionally *the suggested tree representation mainly explains the training set, and not the network mechanism* (in a very complicated way):

(1) Solely using the last layer of the net for the image embedding: Classifiers are known to have a hierarchical structure of attribute representations that are combined to make the classification decision. By looking only at the last layer embedding the author completely disregards that, and the tree representation constructed does not really explain the inner mechanism of the network but only the mechanism of one particular layer.

(2) The refinement stage is made on the training samples; This means that relevant class attribution can be directly extracted by characterizing the leading attribution of the training data, with no connection to the network mechanism itself or any of it embeddings, without any need to synthesize synthetic data.

This stage, combined with the fact that the image representations are extracted only from the last layer, makes the tree representation mainly visualize the hierarchy of the training set not the hierarchy of the operation of the network.


* My second main concern is *the usability of this type of visual explanation*. The author provided some experiments for potential applications but these seem limited; Finetuning classifiers to better obey the representation only marginally improves performance compared to baselines (no baseline performance of the model before fine-tuning was provided). Using the tree for misclassification examination seems only anecdotal with no large-scale explanation or evaluation. Can the trees replace the classifiers and be used to classify images? Is it possible to compose the tree to get a model lever tree?



* Does the explanation tree go beyond the structure of the hierarchy provided in context? I can imagine other important attributes, like the presence of singular objects vs many objects, or different types of conditions like the presence of a dog but only if it is leashed. According to the text, it seems like the “subjects” of each node are fixed according to the in-context example, which significantly limits the flexibility of the representation. Furthermore- how was the format in-context example decided? it seems to have a lot of influence on the results but that was not discussed in the paper.

* In the tree refinement stage, there is no definition for what does it to have “nodes that are seldom visited”, this seems to be an important variable. what is the criterion for that?

* Evaluation-
The faithfulness score seems to directly connect with the way that the tree was defined.
Baseline comparisons are rather ablation study.


* The clarity of text and figures can be improved. Intro is very general and some definitions can be interpreted in many ways, for example, it will be much clearer if the authors state that by in vision model they mean a classifier. Another example is defining the image embedding way before using it which makes it hard to follow details. Also unclear how L_{HMC} is calculated.
The scheme in Fig. 1 was unclear to me. fiuger 5- how were accuracy and MCS calculated here?

**Questions:**

see weaknesses

**Limitations:**

limitations are discussed in the appendix

---

> ### Author Rebuttal · Authors · 2024-08-06
>
> `>>> Q1`**Explaining Network Mechanism**
>
> `>>> A1`Thanks for the question. Actually, our method explains the network mechanism **by identifying prototype sample and concept**.
>
> Unlike *mechanistic interpretability*, which maps concepts to layers or neurons, our LVX uses `prototype-based explanations`[9,10]. We explain the predictions by finding similar samples. Both methods, prototype or mechanistic, explain the network's mechanism, but from different angles.
>
> `>>> Q2`**Last Layer**
>
> `>>> A2`Great question! We focus on the last layer because:
> 1. **Explains Misclassifications:** The last layer's feature directly relate to class probabilities, showing why errors occur.
> 2. **Common Practice:** Using last layer is standard in prototype-based explanation[9,10]. It makes LVX comparable to methods like NBDT[10].
>
> We value the suggestion and will explore more on multi-layer method.
>
> `>>> Q3`**Refinement with Train Set**
>
> `>>> A3`Thanks. Our goal isn't to visualize train set; rather, we use it to quantify how well models understand concepts. Both train set and generated data are essential.
>
> Given generated data with concept `C`, we pass data to models to test their familiarity with `C`. We do this by comparing feature to the train set; high similarity indicates familiarity.
>
> Leading concept in train set isn't always included in the tree. It is included only when the model assign it to generated data.
>
> **Remove TrainSet:**
>
> Instead, we use `average activation magnitude` on generated data as a familiarity measure. Concepts with magnitudes $<\eta$ are pruned. This method, **W/o TrainSet**, is tested on CIFAR-10 using MCS, MSCD score, and average tree depth.
> |MCS/MSCD/Depth|LVX||W/o TrainSet|||
> |-|-|-|-|-|-|
> ||||$\eta=0.01$|$\eta=0.1$|$\eta=0.3$
> |RN50|**31.1/-1.3**/4.2||23.4/-0.3/6.0|26.9/-0.8/3.7|25.3/-0.5/1.4
> |ViT-S|**31.9/-1.7**/4.3||24.2/-0.4/6.2|27.4/-0.9/3.3|26.1/-0.6/1.8
>
> Without train set, setting a threshold is challenging, leading to trees that are too shallow ($\eta=0.3$) or too deep ($\eta=0.01$).
>
> `>>> Q4`**Use Case**
>
> `>>> A4`Grad to see the question. We care a lot about utility.
>
> **1) Finetuning**
>
> Finetuning indeed leads to good improvements.
> 1. **In-domain Results:** The baselines before fine-tune have been shown in `Table 4, NN` and in `Table 5, Baseline`. On CIFAR10/100, accuracy improves by 0.5%–2%.
> 2. **OOD Results:** Fine-tuning improves OOD results **by 5% on ImageNet-A and -S** (`Table 8` Appendix).
>
> **2) Large-scale Evaluations**
>
> As advised, we run experiments to see the models' common issue and difference.
>
> **Exp1: Common Issues**
>
> We examine common errors in the trees. In rebuttal PDF, `Fig 12` lists the top correct and misclassified attributes, while `Fig 13` plots their numbers at each tree depth.
>
> Classifiers show a **coarse bias**. They recognize `Attributes` like *color* well but struggle with `Substances` based on *shape* and *size*. They also handle shallow attributes better than deep, fine-grained ones.
>
> **Exp2: CNN vs Transformer**
>
> We compare CNNs and Transformers through accuracy differences. We use DeiT-B and ConvNeXt-T due to their similar accuracy.
>
> CNNs excel at local patterns like `Attributes`, while Transformers are better on `Environments`, focusing on contexts. This supports findings that **CNN are biased towards textures over shape**[A]. We will include this insights in the revision.
> |Model|Concepts|Substances|Attributes|Environments|
> |-|-|-|-|-|
> |ConvNeXt-T|*23.1%*|*18.9%*|**45.3%**|18.1%|
> |DeiT-B|22.0%|15.6%|35.7%|**26.3%**|
>
> [A]ImageNet-trained CNNs are biased towards texture,ICLR2019
>
> **3) Tree as Classifier**
>
> Yes, we can use trees as a new classifier called **TreeNN**. For each input, we encode it into a feature, which navigates each tree to compute its MSCD score(`Sec 4.1`). The class with the lowest score is selected.
>
> In a CIFAR10 experiment, TreeNN gets 92.7% accuracy with an explainable decision path.
> ||NN|LVX fine-tuned|TreeNN
> |-|-|-|-|
> |RN18|93.1%|**93.6%**|92.7%|
>
> **4) Compose Trees**
>
> Great question! We can merge category-level trees into a model explanation. We remove root nodes and combine common nodes. The resulting tree provides a holistic view of the classifier.
>
> `>>> Q5`**In-context Example**
>
> `>>> A5`Thanks. Yes, LLM can extend the hierarchy beyond the  template. We use a simple template to show the general idea works.
>
> **1) Non-fixed Hierarchy**
>
> Hierarchy vary across classes. For example, `dog` prompt leads the LLM to generate different hierarchy for `train`, focusing on materials, components, and utility.
>
> **2) Flexibility**
>
> The template can be adjusted for complex cases, such as relationship prediction with object interactions. An example of the relation `carrying` is shown in `Fig 14` (rebuttal PDF).
>
> **3) Selection**
>
> We manually write prompts with template from `Line 139` and annotate 1-5 classes as in-context examples. It works well for CIFAR/ImageNet, which feature single objects and clear backgrounds. We'll extend it in future.
>
> `>>> Q6`**Tree Pruning**
>
> `>>> A6`**One** least visited node and its children are pruned, as defined in `Eq 2`.
>
> `>>> Q7`**Faithfulness Score**
>
> `>>> A7`Good question. Yes, we use the same score for routing and evaluation. *SubTree* baseline is an ablation study to build trees **without routing**. It shows feature similarity is a good indicator for faithfulness.
>
> `>>> Q8`**Clarity and Revision**
>
> `>>> A8`Thanks for the suggestions.
>
> **1) Definition**
>
> We'll add definitions for "classifier" and "embedding" earlier in the paper to enhance clarity.
>
> **2) $L_{HMC}$**
>
> $L_{HMC}$ uses supervised contrastive learning for tree labels. Please see [33] `Equation (3)&(4)` for details. We'll explain it in revision.
>
> **3) Fig 1**
>
> `Fig 1` shows a toy example (left) and a diagram (right). It may contain too much information. We'll split it to improve clarity.
>
> **4) Accuracy and MCS**
>
> Definitions of MCS and TK are briefly mentioned in `Line 253`, and full definition in Appendix `Sec I.1`.

---

> ### Author Response · Authors · 2024-08-10
> **Thank the Reviewer for the Constructive Comments**
>
> Dear Reviewer UiCA,
>
> We would like to thank you again for your constructive comments and kind effort in reviewing our submission. Please do let us know if our response has addressed your concerns, and we are more than happy to address any further comments.
>
> Thanks!

---

> > ### Comment · Reviewer_UiCA · 2024-08-11
> >
> > Thank you for your detailed reply. I still wonder how the method is different than constructing the tree based on *only* examining the attributes of images of a given class C in the training set (without feed-forwarding them to the network). Can you please clarify that?

---

> ### Author Response · Authors · 2024-08-12
> **Response to Reviewer UiCA**
>
> Dear Reviewer UiCA,
>
> Thank you for your question. We are truly grateful for your time and the opportunity to clarify our approach.
>
> ------
>
> If our understanding is correct, your main concern seems to be why our LVX needs LLM to refine trees, instead of using pre-existing attributes from the training set directly.
>
> To address this, we provide a brief response in `Q1` and an expanded explanation in `Q2`. Additionally, we illustrate the difference with an example in `Q3` and offer a quantitative comparison in `Q4`.
>
> `>>> Q1`**Quick Answer**
>
> `>>> R1`LVX **does not** merely replicate attributes seen in the training set. Instead, it dynamically adjusts attributes—sometimes *incorporating unseen attributes or excluding existing ones in the training set*—based on the classifier's perception.
>
> `>>> Q2`**Detailed Answer**
>
> `>>> R2`We can best clarify this by comparing two methods that focus solely on *examining training attributes* and explaining how they differ from LVX.
>
> **Method 1: Human-Annotated Attributes.**
> In this method, each class in the dataset comes with detailed, human-annotated attributes. There is no need for further refinement by an LLM; these attributes are organized into trees directly. We retrieve images for each attributes. For a test image, we match it to these generated image like a nearest-neighbor classifier to determine attributes.
>
> **Method 2: LLM-Generated Fixed Attributes.**
> Here, attributes are initially generated by an LLM based solely on the class name. This method represents the LVX in its initial stage, without refinements.
>
> Compared to LVX, those methods differ mainly in three ways:
>
> 1.  **Attributes $\neq$ Explanations:** Unlike `Method 1` and `Method 2`, where attributes are predefined and not influenced by the classifiers, LVX refines these attributes to better explain how the model sees images. This makes LVX not just about predicting attributes but providing model-specific explanations.
> 2.  **Static vs Dynamic:** `Method 1` and `Method 2` involve a finite set of static attributes. LVX, however, can add or remove attributes based on real-time classifier feedback. It enables open-vocabulary capabilities, as detailed in our paper `Line 108`.
> 3.  **Annotation cost:** LVX eliminates the need for annotation required by `Method 1`, reducing costs.
>
> `>>> Q3`**Example: LVX Adds Attributes Not Found in Training Data**
>
> `>>> R3`We provide an example where LVX introduces attributes not originally present in the data. In `Fig 6` the top-right example, the `Hook` is mis-classified as `Nematode`. Actually, all images of `Nematode` in ImageNet are  *black and white 2D microscope images* and do not have 3D attributes like **"Cylindrical"**. However, LVX introduces such non-existent attributes to the attribute tree, which helps explain why such misclassifications occur. This is not achievable for `Method 1`.
>
> `>>> Q4`**Quantitative Comparison**
>
> `>>> R4`We do an experiment to compare LVX with `Method 1` and `Method 2` in terms of faithfulness and plausibility for the generated trees.
>
> -   **Setup**: Since ImageNet and CIFAR do not have human-annotated attributes, we use the CUB-200 dataset for our experiments. This dataset has 312 binary attributes organized into tree structures, which is ideal for `Method 1`. We compare LVX with `Method 1` (human-annotated attributes) and `Method 2` (LLM-generated attributes without refinement) using the ViT-S classifier. We measure faithfulness using the MSCD score and plausibility using the MCS score.
>
> -   **Results**: The table below shows the performance differences. `Method 1` produces trees that align more with human recognition, indicated by a higher MCS score. However, it does not capture the classifier's internal workings as effectively as LVX, which *achieves a significantly lower faithfulness MSCD score*. `Method 2` performs the worst on both metrics.
>
> | Network | CUB-200 (MCS&#8593;/MSCD&#8595;) | | |
> |--|--|--|--|
> | |LVX | `Method 1`| `Method 2`|
> | ViT-S|28.64/**-1.592** | **29.13**/-0.532 | 26.32/-0.478|
>
> ------
>
> Once again, thank the reviewer for the thoughtful feedback and for recognizing the potential in our work. We will incorporate the discussion in our revision.
>
> Please let me know if you need any further clarification.

---

> ### Comment · Reviewer_UiCA · 2024-08-13
>
> Thank you for the additional explanation. I strongly suggest that all these additional explanations will appear in the paper.
> The paper and figures still need some major revisions to improve readability.
> I decided to increase my score to 4.

---

> ### Author Response · Authors · 2024-08-14
>
> We truly thank Reviewer UiCA for raising the score and all the supportive feedbacks. We plan to include all analytic and ablation experiments in the paper and revise the writing to further strengthen our arguments. Here’s a quick plan for our revisions:
>
> `>>> Q5`**Revision Plan**
>
> `>>> R5`
> 1. For the *experimental part*,
>
> - We'll discuss why using training set attributes doesn't improve explanations (`R1-4`);
> - We'll show how removing the training set images worsens results (`A3`) ;
> - We’ll add more analysis and insights using LVX to explain the models (`A4`);
>
> We hope to include these updates in the main paper, as the NeurIPS allow one more page for camera-ready submission.
>
> 2. For the *writing and figure part*, we realize that the current content is overly dense—lots of information and long descriptions have made some figures too small.
>
>      As suggested, we'll streamline these sections, simplify the logic, and increase the font size for figures for better clarity.
>
> -----------
>
> Should you have any further questions or concerns, please let us know and we will try our best to clarify.
>
> Again, big thanks for all the suggestions—they really help us and this paper a lot!

---

### Official Review · Reviewer_U6GY · 2024-07-13

**Soundness:** 2
**Presentation:** 4
**Contribution:** 2
**Rating:** 5
**Confidence:** 5

**Summary:**

This work proposes a method to understand the prediction of an image classification  model using a tree of attributes. The tree of attributes is originally constructed in a text-only by querying gpt for identifying attributes of given concepts. They then use an image-to-test model (or retrieval model) to obtain a set of images corresponding to each attribute. At inference time, they match the input image to each attribute at the current tree level. The path from the root node to the leaf node is meant to interpret the models decision making process.

**Strengths:**

This method works on models that are not just VLM models.  Similar works rely on models being open vocabulary so that the image can be compared to arbitrary attributes described in text. This method first collects the attribute in text and then converts them to image embeddings so that models can be evaluated even if their image embedding space is not aligned with a text embedding space.

The method is clearly described.

**Weaknesses:**

The primary weakness of this work is that the way the tree is constructed is not affected by the image classification model at all but the explanation for the inner working of this model must be a path that exists in the tree. This then presupposes that the tree contains the same reasoning path used by the image classification model which seems to be a large assumption. For example, the model may identify a certain kind of bird by some spurious correlation in the training data. However, as the tree is created by a language model which does not know about this data, this is very unlikely to exist in the tree. However, this method will output some sort of path which is meant to be the explanation even though it is impossible for the tree to produce the correct explanation in this case.

The evaluation metric for measuring the accuracy of the explanation seems poorly defined (see questions sections).

Human eval doesn’t make sense in this case as humans cant evaluate how well a given explanation corresponds to the true decision making process of the model. (they can only measure how well it corresponds to a process they may have used)

Some motivation for this method seems unfounded. For example, the authors state that “Upon observing a dog, we naturally check for the visibility of its tail.” I think this is not correct.

**Questions:**

Faithfulness Score: Can you explain more directly how this score measures the inner workings of the model?

Figure 3 uses the wrong form of “dear”/“deer”

**Limitations:**

yes

---

> ### Author Rebuttal · Authors · 2024-08-06
>
> We sincerely appreciate R-U6GY's thoughtful comments and suggestions. We answer the questions below and will incorporate them into our revised version.
>
> `>>> Q1`**Tree Construction not Affected by Image Model**
>
> `>>> A1`Thank R-U6GY for the insightful comment. In fact, the tree construction is indeed influenced by image models. *They provide valuable feedback to the LLMs to refine the trees*.
>
> As described in the *Tree Refinement Via Refine Prompt* in `Sec 3.1`, the initial tree is updated to better align with the image model’s feature spaces. In this way, LLM and image model cooperatively build those trees; the final tree should accurately reflects the vision model's internal workings.
>
> **What if Image Models Don't Provide Feedback for Refinement?**
>
> We conduct an experiment to show the importance of image models' feedback in tree construction. We introduce a new baseline called **w/o Refine**. In this setup, the initial tree created by the LLM is fixed. We measure the _faithfulness_ using the MSCD score and _plausibility_ using MCS on CIFAR-10 and CIFAR-100 datasets.
>
> |Network | CIFAR-10 (MCS&#8593;/MSCD&#8595;) |  | CIFAR-100 (MCS&#8593;/MSCD&#8595;)| |
> |-|-|-|-|-|
> || *W/o Refine*| **LVX**| *W/o Refine*| **LVX** |
> |ResNet-18 | 27.73/-0.645|**30.24/-0.971** | 23.18/-0.432 |**25.10/-0.574**|
> |ResNet-50| 28.09/-0.822 |**31.09/-1.329**| 23.44/-0.698 | **26.90/-1.170**|
>
> These results show that feedback from the image model makes trees better reflect the classifier's internal representation. They also align more closely with human-annotated decision paths.
>
> `>>> Q2`**Presupposition of Reasoning Path**
>
> `>>> A2`We truly appreciate the question. We do not assume that the initial tree created by the LLM contains the reasoning path. Instead, we make the tree to achieve this goal; we keep refining the tree to make sure it only includes sub-trees that truly represent the reasoning path. The vision model filters out incorrect nodes. This process is achieved through iterative *tree refinement*.
>
> To support our claim, we conducted experiments using **w/o Refine**, as detailed in `Q1`. If we presupposed LLM is sufficient to generate reasoning path and not refining the trees, the explanation's faithfulness and plausibility scores dropped significantly.
>
> `>>> Q3`**Concepts that cannot be generated by LLM**
>
> `>>> A3`Great question. We acknowledge that LLMs may fail to generate hard cases, like spurious correlations. However, we intentionally ensure that the generated explanations match the distribution learned by the LLM. It offers two key benefits:
>
> 1.  **Coherence to Humans.** The trees are more understandable to humans because LLMs are trained on real-world facts.
> 2. **Efficiency.** LVX searches for the best explanations in a vast space of language. Using LLM-generated text narrows this down, helping us find good explanations quickly.
>
> Besides, tree refinement also help to reduce failures. LLMs get to know about the data, through feedbacks from the image model.
>
> We will incorporate this discussion in our revision.
>
> `>>> Q4`**Evaluation Metrics Definition**
>
> `>>> A4`Sorry for the confusion. We have briefly described metrics in the main paper `Sec 4.1` and provided the formal definitions in `Sec I.1` in appendix. As suggested, we will revise these definitions to include more intuitive descriptions.
>
> `>>> Q4`**Human Evaluation**
>
> `>>> A4`Thanks. Human evaluation indeed makes sense, for assessing *plausibility*, not *faithfulness*. It measures how well explanations align with human logic, as mentioned in `Line 613`. Besides, human studies are common method to evaluate explainable methods [6,11].
>
> Faithfulness is what the reviewer referring to. It checks how explanations match the model's decision-making process. For this, we use *Model-induced Sample-Concept Distance* (MSCD), as defined in `Sec 4.1`, with results in `Table 3`.
>
> We evaluate from different perspectives to ensure a thorough evaluation.
>
> `>>> Q5`**Motivation for Prediction-Correction**
>
> `>>> A5`Thanks for the question. Our motivation is founded on **Predictive Coding Theory** [A] in cognitive science.
>
> **Predictive Coding Theory:** The theory suggests that our brain keeps a mental model and makes predictions about what we see. When we see something, the brain compares the actual input with its predictions. If there are differences, or prediction errors, the brain updates its model.  It is what happens both functionally and biologically in the brain.
>
> **Connecting example to Theory:** In the pointed case, we use top-down predictive coding [B]. When we see a `dog`, we expect certain features. If we notice some unexpected features, like `a hairless tail`, our brain updates its understanding of what a `dog` looks like.
>
> As suggested, we will incorporate those cognitive science motivation in our final version.
>
> [A] "Whatever next? Predictive brains, situated agents, and the future of cognitive science." Behavioral and brain sciences
>
> [B] "Predictive coding in the visual cortex: A functional interpretation of some extra-classical receptive-field effects". Nature Neuroscience.
>
> `>>> Q6`**Intuition behind Faithfulness Score**
>
> `>>> A6`Thanks for the question. To be specific, faithfulness score measures **whether the generated tree assigns a test sample to the correct prototypes**.
>
> Given embedding $q_j$ and its explanation $T$, a low faithfulness score $\sum_{v\in V} D(q_j, P_v)$ means the sample is semantically closer to the prototype sets $V$ identified by $T$. It indicates that the network recognizes $q_j$ by assigning it to attributes in $T$.
>
> Conversely, when $D(q_j, P_v)$ is large, it means the explanation fails to recover this assignment and does not reflect the model's decision. We will incorporate this discussion in final version.
>
> `>>> Q7`**Typos Corrections**
>
> `>>> A7`Thank the reviewer for the careful proofreading and we are truly sorry for the typos. As advised, we will fix the them in the revision.

---

> ### Author Response · Authors · 2024-08-10
> **Thank the Reviewer for the Constructive Comments**
>
> Dear Reviewer U6GY,
>
> We would like to thank you again for your constructive comments and kind effort in reviewing our submission. Please do let us know if our response has addressed your concerns, and we are more than happy to address any further comments.
>
> Thanks!

---

> ### Comment · Reviewer_U6GY · 2024-08-12
>
> Thank you to the authors for pointing out that the tree is able to be influenced by the Image Models during the refinement process. I also appreciate the new experiments which demonstrate that this refinement process does lead to a better score.
>
> I also thank the authors for adding citations to support their biological inspiration.
>
> I do however still hold two of my concerns from the previous round of reviews:
>
> 1. Even with the influence of the Vision model during the refinement process, the tree can only be updated by the LLM. This means that the LLM must generate at some point the true reasoning behind the visions models decisions. I think it is not always reasonable to assume that the LLM will do this as the LLM is likely trained on human understanding of concepts which may not always align with the internal working of the model.
> 2. I still do not believe the faithfulness score truly measured the inner workings of the model. Just because the models internal representation of an image is close to the prototype of a given attribute, does not mean that the model is classifying an image a certain way because of that attribute.
>
> I will raise my score to a 4, as the authors addressed a portion of my concerns. However, I believe this work needs more investigation into if the evaluation metric is actually measuring the models reasoning.

---

> ### Author Response · Authors · 2024-08-13
> **Thank the Feedback from Reviewer U6GY**
>
> Dear Reviewer U6GY,
>
> We are sincerely thank you for raising the score and your time to review our work. Besides, we still want to clarify some of the points the reviewer raised.
>
> `>>> Q1`**LLM must Generate at some point the True Reasoning**
>
> `>>> R1`Yes, you are right, we somehow assume LLM can eventually generate true reasoning, even if not in the first round.
>
> While this assumption may have limitations, it offers benefits in terms of **human interpretability** and **efficiency**, as shown in `Q3`. We have also attempted to address potential issues through *tree refinement*, which optimizes the reasoning paths that should be explored.
>
> In addition, this assumption—that explanations can be generated by a model—is common. For example, it is used in counterfactual explanations with diffusion models [A, B] and in training LLMs to generate explanations [C].
>
> We acknowledge that this approach has its limitations and will be explored further in our ongoing research.
>
> **References:**
> - [A] Jeanneret, G., Simon, L., & Jurie, F. Diffusion models for counterfactual explanations. ACCV 2022.
> - [B] Prabhu, Viraj, et al. Lance: Stress-testing visual models by generating language-guided counterfactual images. NeurIPS 2023.
> - [C] Hernandez, Evan, et al. Natural language descriptions of deep visual features. ICLR 2021.
>
> `>>> Q2`**Is the Faithfulness Score Really Faithful?**
>
> `>>> R2`Thanks for the nice question. We believe that measuring feature distance inherently provides a faithful explanation of the model's behavior. This is rooted in how classifiers are trained: Deep classifiers are *trained to perform prototype matching*.
>
> - **Mathematical Rationale:**  Consider a feature extractor $g$ with a linear classifier $W$. Deep networks are trained to minimize the classification error $\text{min}_{g, W} L(W \circ g(x), y)$.
>
>   If we view each column of $W$ as a prototype $W_i$, this objective translates into a prototype matching loss, aligning the sample's embedding $g(x)$ with the prototype $W_y$ of its class $y$: $\text{min}_{g, W} L(W_y, g(x))$
>
>    In this way, classifier make prediction based on its feature distance to its prototypes. In return, measuring this distance provide a faithful way to do explanation.
>
> - **Common Practice:** Despite above reason, using feature distance as a measure in prototype-based explanation is a standard practice in literatures [9,10] .
>
> We understand there are other ways to measure the causality or relevance between model decision and attributes. Using feature distance with prototypes is one of them and, of course, not a very bad one.
>
> -------
>
> Once again, thank the reviewer for the thoughtful feedback and for recognizing the potential of our work.

---

> > ### Comment · Reviewer_U6GY · 2024-08-13
> >
> > I still have some concerns over this faithfulness evaluation metric as it does not seem to me to truly match the process of a classifier. For example, it cannot handle a case where a class either has both attribute A and B or neither (but never only one of these attributes). Essentially, I don't think a mathematical argument for a linear classifier on top of a set of attribute applies to this case (of a non linear classifier on top of a complex uninterpretable set of features).
> >
> > That being said, the authors make a compelling case over the use of this interpretability method in previous works. For this reason I raise my score to a 5.

---

> ### Author Response · Authors · 2024-08-14
> **Thank Reviewer U6GY for the question**
>
> Big thanks to Reviewer U6GY for raising the score and acknowledge our efforts! We appreciate your concern and fully understand that our faithfulness score isn't perfect; it involves certain simplifications, particularly in linear measurements over prototype similarity. It matches the classifier’s decision process, but up to the last layer.
>
> Luckily, despite these simplifications, this score can handle the co-occurrent attributes, in the case you raised.
>
> `>>> Q3`**Faithfulness score and Co-occurrence of Attributes**
>
> `>>> A3`That's a great question. In fact, our faithfulness score can handle the co-occurring attributes. It calculates the sum of feature distances, which can be explained as computing the *negative log-likelihood (NLL) of attributes A and B jointly existing* in the image.
>
> - **Probabilistic Rational:** Consider each attribute as an isotropic Gaussian $P(x|i) \sim \mathcal{N}(\mathbf{p}_i, \mathbf{I})$, with its mean at a prototype embedding $\mathbf{p}_i$. When assuming conditional independence among these Gaussians, the joint probability $P(x|1,\dots,N) = \prod_i^N P(x|i)$.
>
>   By calculating the distance between an embedding $\mathbf{q}$ and these prototypes $\sum_i ||\mathbf{p}_i-\mathbf{q}||^2$, we are actually computing the negative log-likelihood that $\mathbf{q}$ belongs to the joint distribution. This can be expressed as:
>
>     $$\sum_{i}^N ||\mathbf{p}_i-\mathbf{q}||^2  \propto  \sum_i^N -\log(P(\mathbf{q}|i)) = -\log(P(\mathbf{q}|1,\dots,N))$$
>
>     Thus, the feature distance reflects the NLL, showing *how likely the sample contains all the attributes simultaneously*. While these assumptions may somewhat be overly strict, in the probabilistic view, our faithfulness score indeed can measure the co-occurrence of attributes.
>
> - **Joint Attribute Data:** Besides the probabilistic rationale, we also deliberately *collect data containing all attributes within a path from root to leaf*. For example, if the support image contains `a furry tail` for a dog, it will, of course, include `a tail`, because it is an ancestor node in the tree. So, given a test image, if multiple attributes on the same path are identified, they must co-exist in the image.
>
> In both sense, our faithfulness metric becomes a good indicator for co-occurring attributes.
>
> --------------
> Should you have any further questions or concerns, please let us know and we will try our best to clarify.
>
> Again, thank your for all the suggestions and questions this week—they've really helped us to improve our paper!

---

### Author Rebuttal · Authors · 2024-08-06

We would like express our sincere gratitude to all reviewers for their constructive comments. We are particularly thankful for the following positive feedback:

-   The idea is interesting and novel:  `Reviewer UiCA`, `Reviewer kLjQ`;
-   The contributions of the dataset and evaluation metrics are valuable:  `Reviewer kLjQ`, `Reviewer FVKA`;
-   The evaluations are extensive:  `Reviewer kLjQ`;
-  The results outperforms previous methods: `Reviewer FVKA`;
-   The paper is well-written and clearly presented:  `Reviewer U6GY`, `Reviewer kLjQ`.

We will address the specific questions raised by the reviewers in the subsequent sections of the rebuttal.

---

### Decision · Program_Chairs · 2024-09-25

**Decision:**

Accept (poster)

**Comment:**

Originally the paper received mixed reviews, concerns raised included:

1. the loose coupling between how the explanation tree is constructed and the inner workings of the classification model
2. the reliance on only the attributes for the explanation
3. the validity of the metrics
4. general motivation
5. practical value
6. additional baselines needed
7. clarity of presentation
8. insufficient qualitative insights

Following the rebuttal, it seems to the AC that most concerns were addressed as acknowledged by the reviewers, with some concerns remaining on the need for a major revision to improve clarity (rev. UiCA) who raised their score to 4 (still below acceptance). However, as the majority of reviewers recommend acceptance, with UiCA raising their score, the AC believes this paper can still be a valuable contribution to the proceedings. The AC strongly urges the authors to incorporate all the discussions into their revised manuscript, paying special attention to the constructive suggestions of UiCA (as well as all other reviewers ofc).